# Combining ambitious climate policies with efforts to eradicate poverty

Bjoern Soergel [1✉], Elmar Kriegler [1,2], Benjamin Leon Bodirsky [1], Nico Bauer [1], Marian Leimbach [1] & Alexander Popp[1]

Climate change threatens to undermine efforts to eradicate extreme poverty. However, climate policies could impose a financial burden on the global poor through increased energy and food prices. Here, we project poverty rates until 2050 and assess how they are influenced by mitigation policies consistent with the 1.5 °C target. A continuation of historical trends will leave 350 million people globally in extreme poverty by 2030. Without progressive redistribution, climate policies would push an additional 50 million people into poverty. However, redistributing the national carbon pricing revenues domestically as an equal-per-capita climate dividend compensates this policy side effect, even leading to a small net reduction of the global poverty headcount (−6 million). An additional international climate finance scheme enables a substantial poverty reduction globally and also in Sub-Saharan Africa. Combining national redistribution with international climate finance thus provides an important entry point to climate policy in developing countries.

[1] Potsdam Institute for Climate Impact Research, Member of the Leibniz Association, Potsdam, Germany. [2] Universität Potsdam, Potsdam, Germany. ✉email: soergel@pik-potsdam.de

With the adoption of the Paris Agreement and the Sustainable Development Goals (SDGs) an ambitious agenda for mitigating climate change, fostering human development and protecting the biosphere has been set by the international community. Its implementation requires climate policies to go hand in hand with broader sustainable development objectives[1–5].

Arguably one of the most important targets of this agenda is to eradicate extreme poverty as measured by a daily income below the international poverty threshold (SDG 1.1). However, the impacts of unabated climate change could undermine the efforts to eradicate poverty[6]. Negative economic impacts from increased temperatures would affect countries of the Global South more severely[7,8], leading to an increase in global inequality[9]. Within a given country, poorer households are also more vulnerable to climate impacts[10,11].

The importance of eradicating poverty is also explicitly recognized in the Paris Agreement. Notably, ending extreme poverty would only marginally increase the efforts required to meet mitigation targets[12]. Nonetheless, also mitigation policies could have negative side effects for the global poor. At the international level, a uniform carbon price would lead to higher relative policy costs for developing countries[13,14]. Without compensating measures mitigation policies could also hamper progress towards universal access to clean energy[15,16], thus potentially preventing further development and creating a poverty trap[17,18]. Similarly, higher food prices caused by land-based mitigation measures[19–22] could undermine efforts towards a world without hunger[23–25].

Quantifying the poverty implications of climate change and mitigation policies requires capturing the heterogeneity within countries[10,26,27]. Although these distributional effects are of key importance[28–30], so far most integrated assessment models (IAMs)—the major tools for analysing climate policies—do not represent them[27]. At the same time, the existing empirical literature on distributional effects of climate policies within individual countries (e.g.[31]) lacks the global context required for the analysis of mitigation pathways consistent with the climate goals of the Paris Agreement.

Previous studies considering multiple countries have focused on the poverty implications of moderate carbon prices[32,33], but are limited to a static perspective and/or a moderate number of countries[32], or do not include the important effect of land-based mitigation measures on poverty[33]. An analysis of the poverty consequences of the Nationally Determined Contributions (NDCs)[34] has shown relatively moderate effects on global poverty in 2030. By contrast, our study quantifies the consequences of an ambitious, Paris-compatible mitigation pathway for global poverty until mid-century. As such, we provide an assessment of the potential trade-off between climate action (SDG 13) and poverty eradication (SDG 1), and show how it can be overcome.

Based on a mitigation pathway computed with the state-of-the-art IAM framework REMIND-MAgPIE[35], we compute the resulting changes in the income distribution and the effects on national, regional and global poverty rates. We focus on a scenario with burden sharing through internationally differentiated carbon prices. National redistribution policies are funded from the domestic carbon pricing revenue; we highlight the effect of different redistribution schemes on poverty outcomes. We also explore the effects of international climate finance on poverty alleviation.

## Results

**Poverty trends in reference scenarios.** The development of extreme poverty as measured by the international poverty line of 1.90\$/day (PPP 2011) depends strongly on future socioeconomic development. Here we follow the Shared Socioeconomic Pathways (SSPs[36]) in our assumptions for GDP, population and inequality trends. Using the middle-of-the-road pathway SSP2, and in the absence of climate impacts or mitigation policies, we project a continued reduction of extreme poverty. Nonetheless we find that around 350 million people (uncertainty range: 308–411 million) will remain in absolute poverty in 2030, the large majority of them in Sub-Saharan Africa (Fig. 1). Therefore the target to eradicate poverty by 2030 (SDG 1.1) will be missed if socioeconomic development continues in accordance with recent historical trends.

In the SSP1 and SSP5 scenarios with high income growth and decreasing levels of inequality, poverty is reduced at a faster pace. But even under these optimistic socioeconomic assumptions we project around 190 million (SSP5) and 230 million (SSP1) people remaining in extreme poverty in 2030. In the more pessimistic scenarios SSP3 and SSP4 the reduction of poverty slows down, leading to nearly 500 million people in extreme poverty in 2030 in both scenarios. Qualitatively very similar results, but higher overall poverty projections across all SSPs, are also reported by Crespo Cuaresma et al.[37]. Even regardless of the effects of climate change and mitigation policies, these findings mandate substantially increased efforts towards eradicating extreme poverty.

Looking further ahead, we project around 90 million people remaining in extreme poverty by 2050 in the SSP2 reference scenario. The different scenarios span a range of 10–20 million people (SSP5, SSP1) to around 400 million people (SSP3, SSP4). Note, however, that our projections do not include the impacts of unabated climate change on poverty rates, which would likely increase poverty headcounts considerably, especially in the longer term.

**Effects of climate policy and redistribution.** We focus on ambitious mitigation policies consistent with the 1.5 °C target, implemented through a carbon price. The initial price level is differentiated by regions to model a period of staged accession. Developing regions initially face low carbon prices, but converge to the price level of industrialized regions by 2050 (see Methods and Supplementary Fig. 2 for details). To span the range of different mitigation challenges depending on the socioeconomic and technological baseline, we compute mitigation pathways for the three SSPs implemented in the REMIND-MAgPIE framework: the middle-of-the-road pathway SSP2, the fossil-fuel driven development pathway SSP5, and the more sustainable pathway SSP1. We find that in all three scenarios, mitigation policies without associated redistribution policies would lead to an increase in poverty compared to the baseline trend. However, we show that already a progressive redistribution of the national carbon pricing revenues can substantially alleviate or even compensate this policy side effect.

If the revenues are used in a distributionally neutral way, i.e. without changing the level of inequality, richer households accrue a substantial part of the revenues, while low-income households are only partly compensated for their higher expenditures for energy and food. As a result, we project a substantial increase in poverty rates, most prominently in Sub-Saharan Africa, but to a lesser extent also in India, Latin America and South-East Asia (example for SSP2 in Fig. 2a).

If, on the other hand, the carbon pricing revenues are redistributed in a progressive way (implemented as an equal-per-capita climate dividend), the side effects of mitigation policies on poverty are substantially reduced. In almost all countries outside of Sub-Saharan Africa they are even fully compensated, leading to similar poverty rates as in the baseline scenario or a net reduction of poverty (Fig. 2b). The combination of these two

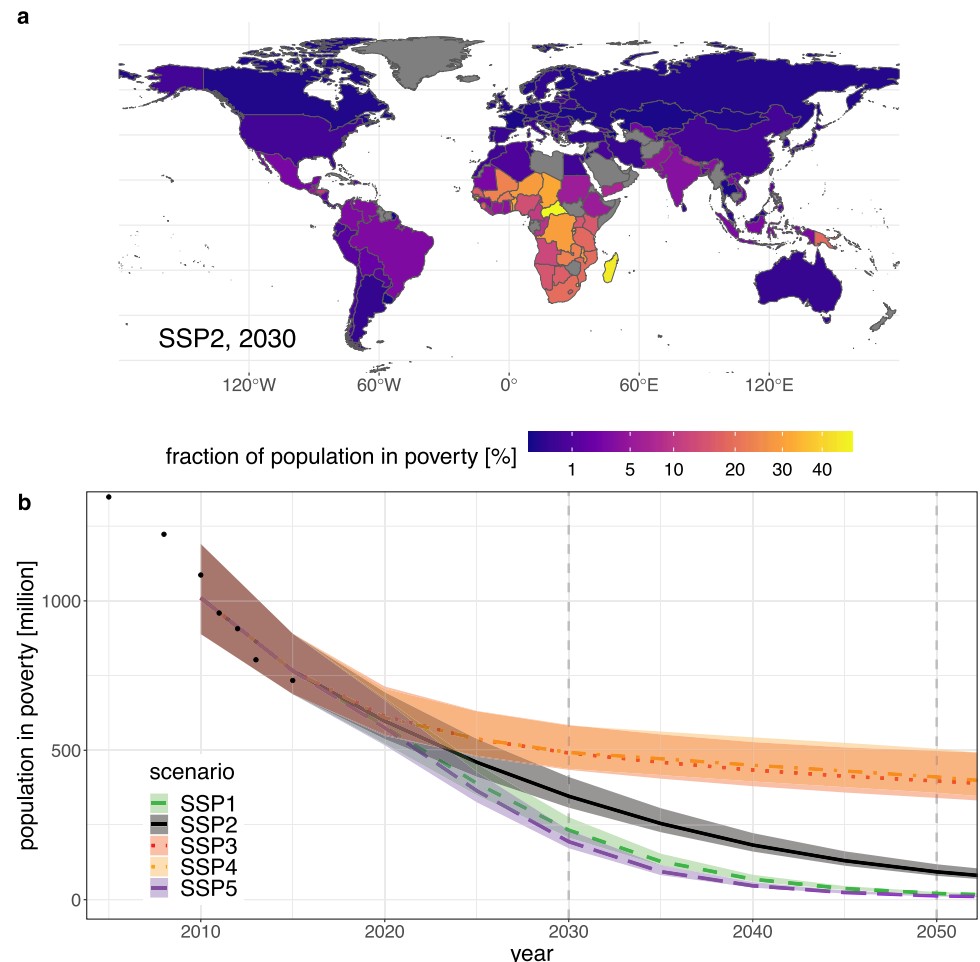

**Fig. 1 National and global poverty trends in reference scenarios with neither climate impacts nor climate policies. a** Projections for national poverty rates in an SSP2 scenario in 2030. For countries that are greyed out in the map, no data was available to calibrate the model for poverty outcomes. **b** Global poverty headcount in the different SSP reference scenarios. While SSP3 and SSP4 are very similar in the global trend, they differ in national and regional poverty projections. The solid/dashed/dotted lines indicate the central projection of our model for the respective SSP scenario. The shaded bands are the 68% prediction intervals; uncertainties are calculated from the regression model for the logit-transformed country-level poverty rates, and propagated to the global poverty headcount (see Section 'Projecting poverty headcounts and uncertainties' in Methods). Recent historical values[73] are shown with black dots.

policies could therefore alleviate or even overcome the trade-off between mitigation of climate change and poverty eradication, thus providing an important entry point to climate policy in developing countries.

Note that in our main analysis we apply this progressive redistribution only to revenues from the energy system, as costs for implementation and monitoring can be expected to absorb a large part of the revenues from pricing land-use emissions. While our distributional analysis and the calculation of poverty rates are performed at the country level, the mitigation pathways are downscaled from coarser regional results (see Methods section for details). As such our results at the national level capture country-specific socioeconomic trends, but do not take into account differences in energy system characteristics or fossil-fuel endowments between countries belonging to the same model region. Therefore we caution against interpreting these results as detailed country-level case studies, and focus on global and regional trends for the remainder of this paper.

**Global poverty headcount.** We show in Fig. 3 the globally aggregated poverty headcount, both for the SSP2 baseline scenario and the SSP2 policy scenario with the two different

redistribution schemes described above. The additional number of people in poverty by 2030 (i.e. the difference between policy and baseline results), both globally and for the four world regions most relevant for the global poverty headcount, is displayed in Fig. 4 for all three SSP mitigation scenarios we consider.

In the case of climate policy without associated progressive redistribution ('neutral') we project an additional 50 million people in extreme poverty in SSP2 by 2030. If, however, the entire domestic carbon pricing revenue is redistributed progressively, the negative side effects of climate policy on poverty eradication can be completely compensated (−6 million people globally). This is an encouraging result: although the implementation of climate policies in developing countries would put a substantial burden especially on the poorest households, already the domestic revenues generated from carbon pricing are sufficient to offset the negative side effect on poverty eradication, at least at the global level. Hence, the reduction of total, national economic income through carbon pricing does not necessarily increase poverty if the revenues are recycled on an equal-per-capita basis (see also Section 7 of Supplementary Information).

This finding also holds for scenarios with different mitigation challenges. In SSP5, mitigation pressure is highest, and thus the

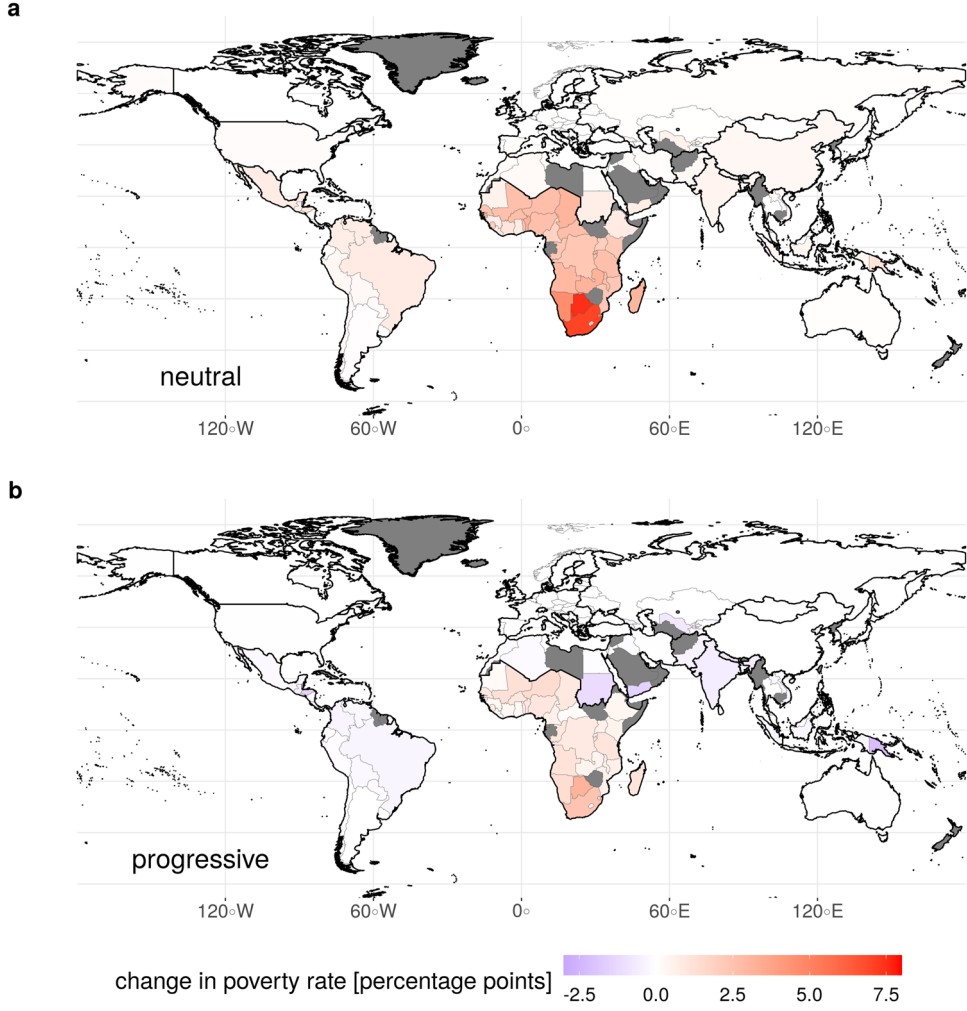

**Fig. 2 Effect of ambitious mitigation policies on poverty rates in 2030 (SSP2).** We show here our projections for the difference between policy and baseline poverty rates for the two different revenue recycling schemes. While climate policy without associated progressive redistribution (panel **a**, 'neutral') would lead to a substantial increase in poverty rates, this policy side effect could be reduced or largely overcome by redistributing the associated carbon pricing revenue (panel **b**, 'progressive'). Light grey lines show national borders, while solid black lines delineate the model regions used in REMIND-MAgPIE.

increase in poverty caused by mitigation policies is comparable to SSP2 despite the much lower baseline poverty. At the same time, also the carbon pricing revenues are highest in SSP5, such that again the policy side effect can be compensated from the revenues. In SSP1, on the other hand, mitigation pressure is lower, and thus also the poverty increase caused by mitigation policies without redistribution is smaller. Again a full compensation of the poverty side effects is possible through progressive redistribution, leading to similar results across the three SSPs.

However, much of the heterogeneity between different regions and countries is lost when aggregating to these global figures, so that the total global headcount does not reflect potential hardships that are regionally concentrated. We therefore also discuss a regional breakdown of our results below.

**Regional poverty trends**. Countries of the Sub-Saharan African (SSA) region have the highest poverty rates today, and also in our projection for 2030 (Fig. 1). In addition, the increases in energy and particularly food expenditures triggered by carbon pricing are substantial (Supplementary Fig. 3). At the same time, the revenues from carbon pricing are modest, both due to the low per-capita emissions from the energy sector and the initially low

carbon price. We thus find that climate policy without progressive redistribution policies would increase the poverty headcount in SSA substantially, by around 30 million in the SSP2 mitigation scenario without progressive redistribution (Fig. 4).

Most of this increase in poverty can be compensated through a progressive redistribution of the carbon pricing revenue, but even under this optimistic assumption there would be an increase in poverty by around 10 million people by 2030. Varying the socioeconomic baseline, we obtain similar trends as discussed above for the global headcount. Notably, however, in SSP1 a near-complete compensation of the policy side effect is also possible in SSA. This highlights that a generally more sustainable development pathway also reduces or avoids potential adverse side-effects of climate policies.

For India we project a fairly rapid reduction of poverty in the baseline scenario, in accordance with projections by the World Bank[38]. Against this background also the effects of climate policy are less severe (+7 million people in SSP2 by 2030 without progressive redistribution). In addition, also the carbon pricing revenues are higher than in SSA, and thus we project that poverty in India could even decrease if climate policy is implemented together with an equal-per-capita redistribution of the revenue (−10 million people by 2030). Interestingly, the poverty reduction

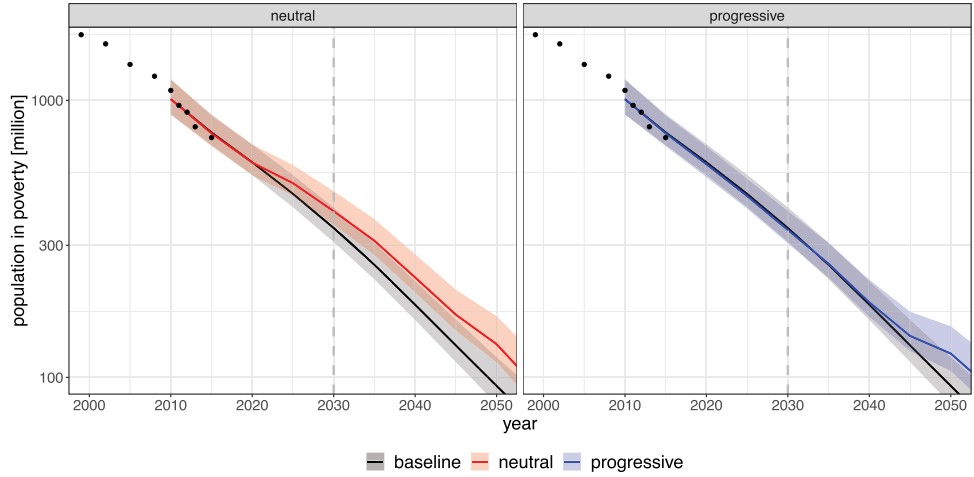

**Fig. 3 Global poverty headcounts under ambitious climate policy (SSP2).** We show here our projections for the global number of people below the absolute poverty threshold of 1.90 $/day (note the logarithmic scale). The SSP2 baseline (black line) is identical to Fig. 1. For the climate policy scenario the poverty outcomes depend substantially on how the carbon pricing revenue is used (coloured lines) — leading either to a slowdown of poverty reduction or to a trend comparable to the baseline. The solid line represents the central projection of our model; the shaded bands are the 68% prediction intervals (see also caption of Fig. 1).

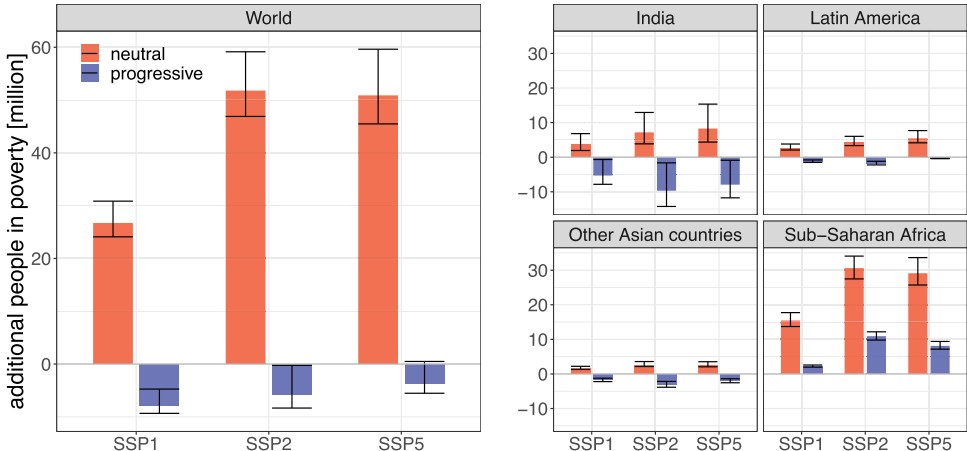

**Fig. 4 Additional number of people in poverty in 2030 (multiple SSPs).** Both at the global level, and for the four most relevant regions, we show the difference between policy and baseline results for SSP1, SSP2 and SSP5. Again we differentiate between the two revenue recycling scenarios. The bars represent the central projection of our model; error bars are the 68% prediction intervals for the difference between the respective policy scenario and the baseline case (see Section 'Projecting poverty headcounts and uncertainties' in Methods).

achievable through progressive redistribution is lowest in SSP1, reflecting the already low baseline poverty and the (compared to SSP5) modest carbon pricing revenues.

In the other Asian countries (not including China, which is a separate model region) and in Latin America the trends are qualitatively comparable to India, but their contribution to the global numbers is smaller. Overall, our regional analysis reveals a strong heterogeneity in the ability of countries to compensate the distributional side effects of mitigation policies from their domestic carbon price revenue. While for most countries the revenues are sufficiently large to avoid an increase in poverty at least in the near term, this is not the case in Sub-Saharan Africa.

**Options for generating poverty co-benefits.** So far we have focused on the question whether an equal-per-capita redistribution of the carbon pricing revenues is sufficient to avoid poverty side effects of ambitious mitigation policies. We now investigate if it is possible to achieve a poverty co-benefit, i.e. a net reduction of poverty through a combination of climate policy and

redistribution measures. One option to achieve this would be to redistribute the national revenues in a strongly progressive way to maximize their effect for poverty reduction. In addition, we explore two ways to increase the revenue base available for redistribution especially in developing countries, an inclusion of the revenues from pricing land-use related emissions, and an international climate finance mechanism funded from a fraction of the carbon pricing revenues.

Here we explore these three options for the SSP2 mitigation scenario, and show the results in Fig. 5. Note that we also include results for 2050 into this assessment. In our ambitious mitigation scenarios, $CO_2$-neutrality is achieved around this time. Therefore the small remaining carbon pricing revenues (Supplementary Fig. 3) in our default scenarios are insufficient to compensate the remaining policy side effects (around +30 million people in poverty in SSP2 after progressive redistribution).

*More progressive redistribution.* We implement a strongly progressive redistribution scheme as a redistribution inversely proportional to income. We find that redistributing the carbon pricing

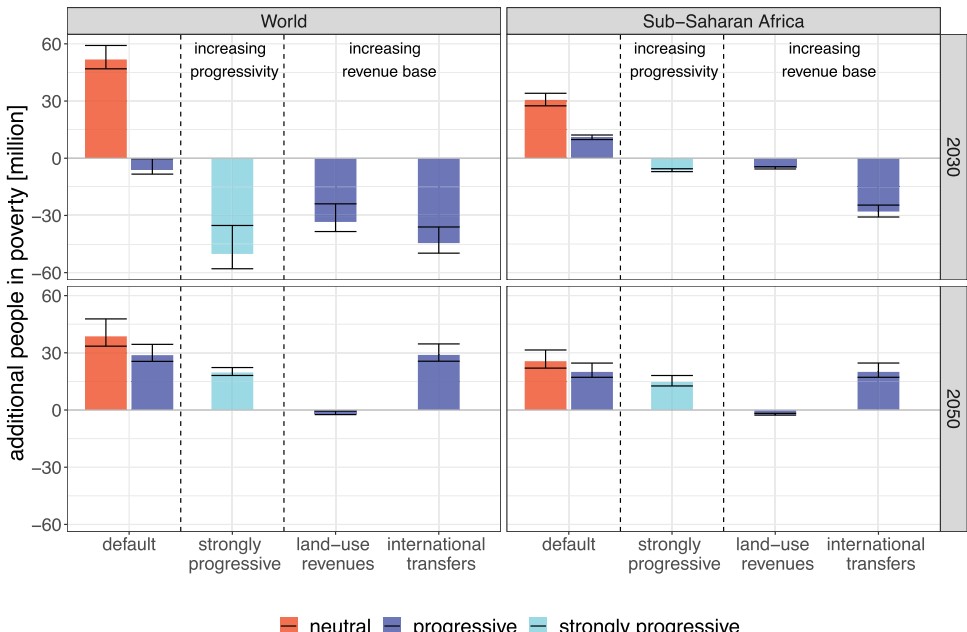

**Fig. 5 Additional measures to achieve poverty co-benefits (SSP2).** We show three different ways to enhance the poverty reduction co-benefits of the revenue recycling. The first option has the same revenue base as the default case (shown at the very left for comparison), but redistributes them in a strongly progressive way. The second and third options, land-use revenues and international transfers, increase the revenue base available for redistribution. Left and right panels are global results and Sub-Saharan Africa, respectively; top and bottom panels 2030 and 2050. Bars represent the central projection, error bars are 68% prediction intervals (see also caption of Fig. 4).

revenues according to this scheme is able to fully compensate the poverty side effects of mitigation policies in SSA in 2030 (Fig. 5). As a consequence also the global poverty headcount is reduced to significantly below the baseline value (−50 million people). In 2050, on the other hand, even a strongly progressive redistribution cannot compensate the policy side effects, as the remaining carbon pricing revenues available for redistribution are small.

However, it is unclear whether such a scheme could be implemented in practice in most developing countries, how effective its targeting would be, and how much of the revenues would be absorbed by the cost of administering the scheme (see e.g. the discussion in Banerjee et al.[39]).

*Redistribution of revenues from land-use emissions.* For our main results we have assumed that only carbon pricing revenues from the energy system are available for progressive redistribution policies, as pricing land-use emissions (in particular methane and nitrous oxide) likely comes with larger costs for implementation and monitoring. If, however, a pricing of these emissions from agriculture and other land uses could be achieved at modest transaction costs, it would increase the revenues available for redistribution policies (Supplementary Fig. 3). This is especially important for Sub-Saharan Africa, as a substantial part of the greenhouse gas (GHG) emissions in these countries originate from this sector. We find that the additional revenues—if redistributed progressively—are sufficient to compensate the increase in poverty in all countries of SSA, both in 2030 and 2050 (Fig. 5). As a result, we project global poverty figures to be reduced below the baseline value in 2030 (around 30 million people less), and to be nearly identical to the baseline results in 2050. While it is unclear if such a scheme could be implemented in the near term, it might be a useful measure for compensating residual side effects of mitigation policies as $CO_2$ neutrality is approached.

*International climate finance.* In our main analysis we have assumed that international burden sharing is implemented

through a period of staged accession, where developing regions face substantially lower carbon prices than industrialized regions until 2050. In addition, we now implement an international climate finance mechanism in a stylized way by transferring 5% of the energy-sector carbon pricing revenues from the industrialized countries to the Sub-Saharan African countries, where they are redistributed alongside the domestic revenues. This implies international transfers of initially around 100 billion $/yr (around 0.2% of GDP of the donor countries), but decreasing towards mid-century as emissions are reduced. Note that this level of climate finance mirrors the commitment by industrialized countries during the UNFCCC negotiations and in the Paris Agreement[40].

In combination with an equal-per capita redistribution scheme, such a mechanism would even lead to lower 2030 poverty rates than in the baseline scenario (− 30 million people in SSA, − 45 million people globally). Already modest international transfers funded from a fraction of the carbon pricing revenues of industrialized countries are thus sufficient to overcome the residual trade-off between SDG 1 and SDG 13. Such a mechanism would also align with SDG 17 ('partnership for the goals', in particular target 17.2).

By 2050, on the other hand, carbon neutrality has largely been achieved in the industrialized countries, therefore we assume that also the climate finance transfers cease to exist. As a result, we again project an increase in poverty (+ 20 million people in SSA, + 30 million people globally). Therefore additional funds beyond the transfer of carbon pricing revenues would have to be sourced.

A concern related to international transfer mechanisms is that the financial inflows might have negative consequences for the economies of developing countries, potentially leading to a 'climate finance curse'[17]. However, our stylized scheme implies international transfers well below the ones suggested by common burden sharing schemes[14,41]. They are also well below the (largely unfulfilled) target of many developed countries to provide at least 0.7% of gross national income as official development assistance.

**Sensitivity to mitigation target**. It is well known that policy costs increase non-linearly with the stringency of the temperature target[42], and especially so for low-income countries[41]. As such, also larger poverty side effects can be expected, but at the same time there are higher carbon pricing revenues available for redistribution policies. Repeating our analysis with less stringent mitigation targets corresponding to a well-below 2 °C and a 2 °C temperature target, we find that the net poverty outcome after progressive redistribution of the revenues worsens slightly for more lenient temperature targets. For the case of the 2 °C temperature target, a small net increase in poverty remains (+5 million people globally in 2030; see Section 10 of SI and Supplementary Fig. 10). This reflects that ambitious targets lead to higher carbon pricing revenues in the near term, which, however, diminish more rapidly over time as $CO_2$ neutrality is approached faster.

**Higher poverty line and longer-term prospects for poverty eradication**. So far we have focused on the international poverty line of 1.90 $/day, and mostly on the 2030 horizon set by the SDGs. However, the international poverty line is also criticized as being too low for acceptable living standards in many countries (e.g.[43–45]). We thus repeat our analysis with a higher poverty line of 5.50 $/day, which is motivated by the value currently used by the World Bank for upper-middle income countries. We focus on this higher poverty line when analysing the longer-term poverty trends until 2050.

We project that poverty figures as measured by this higher poverty line will remain high until mid-century, especially in Sub-Saharan Africa, but to a lesser extent also in India and certain countries of Latin America and Asia (Fig. 6a). For the 'middle-of-the-road' SSP2 baseline we project a global poverty headcount of around 2.5 billion in 2030, and around 1.4 billion in 2050 (again without the additional effects of climate impacts). The other SSPs span a range between 280 million (SSP5) and 3.2 billion (SSP3) in 2050, with the latter being close to the current value. Again we find that at the global level SSP1 and SSP5, as well as SSP3 and SSP4, respectively, have broadly comparable poverty trends (Fig. 6b).

Against this background of high baseline poverty rates, we also obtain substantial side effects of climate policy, which persist also after the redistribution of the small residual carbon pricing revenues (e.g. +200 million for SSP2 with equal-per-capita redistribution in 2050, Fig. 6c). Again, this policy side effect is much less pronounced in SSP1, reinforcing our earlier finding that a more sustainable development pathway not only makes mitigation targets easier to achieve, but also reduces the side effects of climate policies. Out of the previously discussed additional measures, only a redistribution of revenues from pricing land-use emissions is able to largely compensate policy side effects on poverty (Fig. 6d), demonstrating its value for longer-term poverty eradication.

Taken together, this analysis of longer-term trends highlights that eradicating poverty, and avoiding adverse side effects of climate policies, requires also looking beyond the 2030 horizon given by the SDGs. Substantially increased efforts towards poverty eradication are thus mandated, especially when considering a higher poverty line that goes beyond the 1.90 $/day definition of extreme poverty.

## Discussion
Poverty outcomes depend on the distribution of mitigation efforts between countries and over time, between sectors and income groups within countries, and on the use of the carbon pricing revenue. To our knowledge, our study is the first to capture all of these layers at least to some extent. Our main finding is that there are substantial side effects of mitigation policies on poverty eradication, but already a progressive redistribution of the national carbon pricing revenues is sufficient to largely compensate for them.

Nonetheless average per-capita income levels in developing countries would decrease considerably under ambitious mitigation policies (Supplementary Fig. 6), as even under our strongly differentiated carbon prices the international distribution of mitigation costs is regressive. Therefore there is still a need for an equitable international burden sharing. Indeed we find that already modest international climate finance transfers would lead to a substantial reduction in near-term poverty headcounts.

We aimed for a poverty analysis with global coverage until mid-century, which is only possible by taking a fairly aggregate perspective: we only use national statistics for the distribution of income, and do not distinguish between different final energy carriers or food commodities. A greater level of detail in the incidence of policy costs could be achieved with an input-output approach[12,33,46]. Sectoral poverty dynamics could be disentangled with a computable general equilibrium (CGE) setup connected to detailed household surveys[32]; both of these techniques are however often limited to a static perspective.

Using the latter approach, Hussein et al.[32] find that ambitious mitigation policies in developing countries would increase poverty rates, similarly to our results without progressive redistribution. Campagnolo & Davide[34] also employed a CGE model, but use its results (e.g. public education expenditure, sectoral value added, unemployment) to drive regression models for inequality and poverty. While this models the effects of climate policy on poverty via these structural variables, it does not fully capture the direct distributional effects through energy and food prices and revenue recycling.

Our analysis has focused on these direct distributional effects but did not include other potentially heterogeneous effects of climate policies (e.g. employment loss/creation). Furthermore, we have not captured the increased income for agricultural households when food prices rise[20,47,48]. In our setting the main drivers for food price increases are emission pricing and land scarcity driven by land-based mitigation options. The former does not result in additional income for farmers, and it is unlikely that increased returns to land ownership would substantially benefit poor households[25,32]. Nonetheless, a more detailed coverage of potentially heterogeneous effects on the income side would be a valuable extension for future work. Should such a quantification become available, it can be incorporated into our framework by specification of the appropriate income elasticity.

Our progressive redistribution policies require sufficient institutional capacity, and we assume that there are no substantial transaction costs. The latter seems reasonable in the case of lump-sum transfers, but transaction costs could increase if regular small payments were made instead. Requirements for institutional capacity and costs for administering the policy would likely also increase for schemes that are more progressive than an equal-per-capita redistribution.

Instead of directly redistributing the carbon pricing revenues, governments could also use them to increase their spending for other poverty-reducing policies. This includes for example education spending, but also infrastructure development that is critical for achieving other SDGs, such as access to electricity, clean water, sanitation, transport and telecommunication[49–51]. While we do not attempt to directly quantify the effect of such policies, our different redistribution schemes can be seen as stylized explorations of different degrees of progressivity in spending the revenues from carbon pricing.

We reiterate that we did not include the effect of climate impacts, but focused on quantifying the poverty side effects of

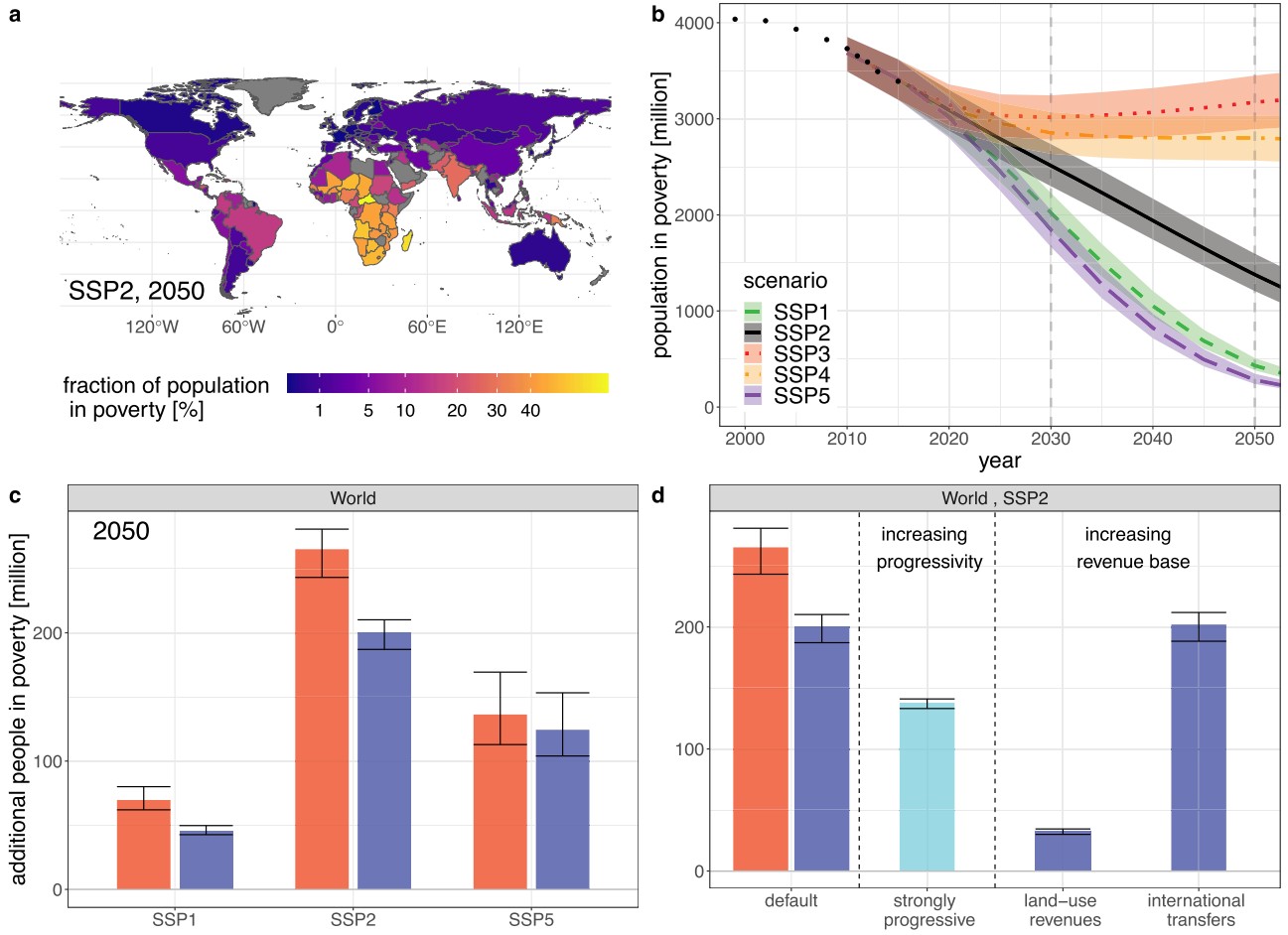

**Fig. 6 Longer-term prospects for poverty eradication, using a higher poverty line of 5.50 $/day. a** National poverty rates in 2050 (SSP2). **b** Global poverty trends (all SSP baselines). See the caption of Fig. 1b for a definition of central estimate and uncertainty bands. **c** Longer-term effects of mitigation policies on poverty eradication (multiple SSPs, 2050). See Fig. 5 for the color legend and definition of error bars of panels (**c** and **d**). **d** Effect of additional policies (SSP2, 2050).

mitigation policies. At the same time, ambitious mitigation measures are of particular importance for the global poor, as they are most vulnerable to the impacts of climate change[6,8,10]. We further note that the effects of the COVID-19 pandemic on poverty eradication[52,53] are not captured in our projections, as the effects of the pandemic on mid- to long-term economic development are not yet clear.

Of course the effects of mitigation policies on developing countries are also broader than the one-dimensional metric of extreme poverty[17,54,55]. Including climate impacts, including the effects of the COVID-19 pandemic, and extending our approach to additional SDG dimensions are therefore high priorities for future research.

Such a quantitative and multi-dimensional assessment of SDG outcomes would substantially enhance the value of IAM scenarios in navigating trade-offs between different policy objectives. In particular, it would ensure that the policy recommendations emerging from studies of mitigation pathways avoid putting a disproportionate burden on the global poor. The latter is a necessary condition for jointly implementing the Paris Agreement and the SDG agenda.

## Methods

**Overview of methodology**. We calculate poverty rates as a post-processing of a mitigation scenario computed with the IAM framework REMIND-MAgPIE[35,56,57]. The key steps of our analysis are evaluating the changes to the income distribution as a result of mitigation policies, and linking these changes to poverty outcomes.

A flow chart summarizing the different analysis steps is provided in Supplementary Fig. 1.

Here we first give a brief overview of our IAM framework and explain the scenario setup chosen for this work. Subsequently, we discuss how to compute the components required for our post-processing analysis from the IAM output, in particular the GDP loss, additional energy and food expenditures, and the carbon pricing revenue. In our newly developed distributional framework we distribute these to different income groups. This allows us to compute an income equivalent net of climate policy induced changes, as well as the corresponding Gini coefficients, in ambitious mitigation scenarios at the country level. We translate these into national poverty rates using a regression model calibrated on recent poverty and inequality data. Finally, we calculate national, regional and global poverty headcounts which form the main result of our analysis. Detailed explanations and derivations of many analysis steps are available in the SI.

**REMIND-MAgPIE IAM framework**. REMIND-MAgPIE is an IAM framework coupling a global, multi-regional energy-economy-climate model (REMIND) to a spatially-explicit land-system model (MAgPIE). A description of the most salient features of the two individual models is given in the SI; they are also documented in detail in refs. [58–61] and references therein. The source code and documentation for both models are available online (link in SI).

The energy-economy and land-use systems are connected as follows: the carbon price from REMIND is applied to land-use-related emissions in MAgPIE, which are in turn also taken into account in REMIND. In addition, the availability of bioenergy as a mitigation technology links the two systems. The coupled system REMIND-MAgPIE is run in an iterative way: information about the $CO_2$ price, emissions and bioenergy demand and price are exchanged until a joint equilibrium is reached[35,56,57].

**Scenario description**. Our assumptions for socioeconomic development follow the Shared Socioeconomic Pathways (SSPs[36]). For the three SSPs implemented in

REMIND-MAgPIE (SSP1, SSP2, SSP5), we compare a baseline scenario (without climate policy or climate impacts) to an ambitious mitigation scenario that limits the increase in global mean temperature to 1.5 °C. Importantly, this comparison should not be misinterpreted as evaluating the trade-off between poverty in a world with unabated climate change, and poverty caused by mitigation measures. Instead, our baseline scenario forms a counterfactual reference case for socioeconomic development in the absence of climate change and mitigation measures (as do the SSPs in general). Our baseline therefore does not include the (very likely substantial) effects of climate impacts on poverty.

Our mitigation scenarios are implemented as a regionally differentiated carbon tax, which is adjusted endogenously such that a $CO_2$ budget of 900Gt from 2011 until the (endogenously determined) time of peak warming[62,63] is not exceeded. Climate policy starts after 2020 with a period of staged accession: in developed economies the $CO_2$ price increases steeply until the peak budget is reached, and flattens off afterwards. Developing countries, on the other hand, initially face a lower carbon price that converges to a globally uniform price by 2050 (Section 2 of SI). Our level of price differentiation goes significantly beyond what is assumed for the period of initial fragmentation in the Shared Policy Assumptions[64], but is lower than what would be needed to equalize fractional mitigation costs[41].

The carbon price levels required to meet the 1.5 °C target are determined endogenously as part of the REMIND-MAgPIE optimization and are shown in Supplementary Fig. 2. We find that for SSP2 the resulting carbon prices in 2030 would have to be around 330$ [USD 2005] in industrialized economies. On the other hand, Sub-Saharan-Africa would face a much lower (but still substantial) price of around 55$ in 2030. Carbon prices in our SSP1 mitigation scenario are similar to SSP2, reflecting a compensation between lower energy demands on the one hand, and a more restricted technology portfolio (e.g. limits on carbon capture and storage) on the other hand. Due to the high energy demand in SSP5, carbon prices would have to be around 50% higher than in SSP2.

While our IAM model runs use a time horizon until 2100, here we only discuss results until 2050, as in particular the projections for within-country inequality become increasingly uncertain for longer time horizons.

**Policy cost metrics.** Our distributional calculation is performed as a post-processing of the IAM runs. As a metric for the effect of climate policies on household incomes, we calculate the income equivalent net of climate policy induced changes. Importantly, this takes into account both the income and the expenditure side: when energy and food prices rise as a consequence of mitigation policies, individuals can purchase less of those goods with their income, making them poorer in real terms[18,32,65]. (Note that we also include price-induced changes in energy and food quantities; see Sec. 3.1 of SI for details). Equivalently, this can be viewed as evaluating the total welfare change from both the income and expenditure side using a monetary metric of utility[66]. We track the following components, which we calculate from the difference of mitigation and respective baseline scenario:

- GDP loss: we take this as an aggregate measure of total income loss, as our IAM, unlike a CGE, does not provide a high sectoral resolution of the income side.
- additional expenditures for final energy (FE)
- additional expenditures for food
- redistribution of the net GHG pricing revenue (see Section 3.2 of SI for details)

We express these components as a fraction of GDP in the baseline scenario, and denote them by $\delta^{GDP}$, $\delta^{FE}$, $\delta^{food}$ and $\delta^{GHG}$ respectively. A detailed description of how these components are computed is given in the SI; Supplementary Fig. 3 shows an overview of the resulting values. Note that we apply a rescaling of the prices for food commodities computed in MAgPIE to make them more representative of the prices that households in developing countries are confronted with (see Section 3.1 of SI).

In our main analysis we use the revenues from pricing $CO_2$, $CH_4$ and $N_2O$ from fossil fuel use and industry emissions for direct redistribution policies. We also discuss how an inclusion of land-use related $CH_4$ and $N_2O$ emissions into the redistribution scheme would change poverty outcomes. The same holds true for an international transfer scheme that uses a fraction (5%) of the carbon pricing revenues from industrialized regions to offset policy side effects in developing countries (Section 3.2 of SI). We assume that these climate finance funds are transferred to the Sub-Saharan-African countries, and redistributed alongside the domestic revenues raised there. We focus the international transfer payments on this region, as our results show that most other countries are able to compensate the poverty side-effects from their domestic revenues.

As our distributional calculation is performed at the level of individual countries, we downscale the REMIND-MAgPIE results by assuming equal fractional costs (GDP loss, increased food and energy expenditures) and carbon pricing revenues for all countries belonging to the same model region. In other words, we assume that price increases, associated demand responses, macro-economic effects etc. are comparable for all countries within a region. This downscaling step is necessary for an analysis with global scope, as it is computationally not feasible to compute country-specific mitigation pathways while maintaining global coverage.

**General distributional framework.** In our distributional framework we start from a baseline income distribution and baseline price levels, and subsequently calculate the changes caused by mitigation policies using the four policy cost metrics $\delta^j$ ($j = \{GDP, FE, food, GHG\}$) computed from the IAM output.

We assume that the average per-capita income in every country is given by the GDP/capita values for the respective SSP[67]; the level of intra-national inequality is determined by the Gini projections by Rao et al.[68]. (Note that we harmonize the SSP Gini coefficients to the SSP2 values until 2020 to avoid divergence in the scenarios already in the historical period. The values we use for SSP{1,3,4,5} in our projections are thus shifted by the (mostly small) difference to SSP2 in 2020 compared to the original Gini coefficients by Rao et al.) Based on these inputs we model the baseline distribution of income in every country as $y \sim \text{Lognormal}(\mu, \sigma)$; see Section 4 of the SI for details.

The loss (or gain) due to policy cost category $j$ for a person with baseline income $y$ is then given by

$$\Delta y^j = \delta^j\, \bar{y} \times y^{\alpha_j} e^{-\alpha_j \mu - \alpha_j^2 \sigma^2 / 2};\qquad(1)$$

this is derived from the initial lognormal distribution by requiring that losses are proportional to $y^{\alpha_j}$ while ensuring that the national average is preserved (see Sec. 4 of the SI for details). Here $\bar{y}$ is the average per-capita income in the baseline scenario and $\alpha_j$ is the income elasticity of mitigation costs for category $j$, which quantifies how the aggregate national costs are distributed. For example, $\alpha_j = 1$ results in an equal relative income loss for all individuals, whereas $\alpha_j = 0$ would imply the same absolute income loss and thus a highly regressive distribution of costs.

Eq. (1) forms the core of our distributional analysis. We now apply it to the different categories of changes to the income distribution calculated from the IAM output. A distinctive feature of this approach is that we model the changes in the entire income distribution, as opposed to existing approaches that only work with a small number of income groups (typically quintiles or deciles).

**Distribution of policy costs and revenues.** We decompose the total change in income equivalent (or welfare measured in monetary units) from climate policy into an income side and an expenditure side: average incomes are reduced, and households pay higher prices for food and energy (note that our final energy and food prices include the carbon price). At the same time, they can benefit from a redistribution of the revenues from carbon pricing. The income equivalent of an individual with baseline income $y$ therefore changes to

$$\tilde{y} = y - \Delta y^{GDP} - \Delta y^{FE} - \Delta y^{food} + \Delta y^{GHG};\qquad(2)$$

note that in our sign convention all $\Delta^j$ terms are positive. We calculate individual policy costs for each category as follows (see Section 5 of SI for details):

- Overall GDP loss is assumed to be distributionally neutral ($\alpha_{GDP} = 1$).
- Increased energy expenditures are distributed according to income-dependent final energy expenditure shares, reflecting the empirical finding that in low-income countries the energy expenditure share increases with growing income, whereas the opposite is true for higher-income countries[33]. We estimate the corresponding income elasticity of final energy expenditures, $\alpha_{FE}$, empirically from the data Dorband et al.[33] compiled from the World Bank's Global Consumption Database[69]. This data set provides energy, food, and total expenditures at the level of four consumption groups per country for a large number of countries. The income elasticity relates to the final energy expenditure share computed from the survey data as

$$\alpha_{FE} - 1 = \frac{d \log (\text{FE exp. share})}{d \log y}\ .\qquad(3)$$

A detailed description of our empirical method is given in Section 5.2 of the SI.
- For additional food expenditures we apply the same procedure as for energy. The resulting income elasticity of food expenditures, $\alpha_{food}$, reflects empirical findings that food expenditure shares decrease substantially with increasing per-capita income levels[70].

Importantly, our distributional analysis also makes the effects of different redistribution schemes explicit. In contrast, standard IAM analyses with only one representative household per model region implicitly assume 'perfect' redistribution within every region, such that neither climate impacts nor policy costs change the level of inequality[71]. Here we implement three different schemes for the redistribution of the carbon pricing revenue:

- The carbon pricing revenue is spent in a distributionally neutral way, i.e. changing the average income but not the level of inequality. Technically this is implemented as a redistribution proportional to income, i.e. $\Delta y^{GHG} = \delta^{GHG} y$. This approximates a case where the carbon pricing revenue is used to reduce other taxes, but not in a progressive way.
- The revenue is used to fund a progressive redistribution, implemented as an equal per-capita payment for all individuals: $\Delta y^{GHG} = \delta^{GHG} \bar{y}$. This represents an optimistic scenario where all countries commit to redistribution policies alongside their climate policy. Note that this assumes functioning institutions, such that there are no substantial leakages or inefficiencies in the redistribution scheme.

- To assess the potential co-benefits of strongly progressive redistribution policies funded from the carbon pricing revenue, we explore a scheme where redistribution is inversely proportional to income, i.e. $\alpha_{GHG} = -1$. Note that such a scheme would have even larger requirements for institutional capacity than discussed for the equal-per-capita case above.

**Monte Carlo simulation for new distribution.** Subtracting additional expenditures and adding transfers from carbon pricing revenue changes the income distribution away from the initial lognormal case. We calculate the distribution in the climate policy scenario numerically with a Monte Carlo simulation, which models the population of a country in a given year with one million representative individuals. From these samples, we readily compute the average income and the Gini coefficient in the policy scenario for every country (see Section 6 of SI for details), which we use in the subsequent poverty analysis. We note, however, that also any other desired summary statistic (such as other inequality metrics, e.g. the Palma ratio[72]) can be inferred from our method. As an intermediate result of our distributional analysis, we show and discuss average incomes and Gini coeffients for four representative countries in Section 7 of the SI.

**Regression model for poverty outcomes.** In the above we have computed changes to the income distribution through climate policy and the associated redistribution policies. We now connect these changes to poverty outcomes using a regression model with average income ($\bar{y}$) and Gini coefficient as main drivers. For our main analysis we define poverty following the international poverty line of 1.90\$/day in 2011 PPP dollars, i.e. we assume a constant poverty line in real terms. In addition, we explore a higher poverty line of 5.50 \$/day, especially for analysing longer-term poverty trends.

Denoting the share of the population in country $c$ at time $t$ that is above the poverty line by $s_{c,t}$, our regression model is specified by

$$\log \frac{s_{c,t}}{1 - s_{c,t}} = \beta_0 + \beta_1 \log \bar{y}_{c,t} + \beta_2 G_{c,t} \\ + \beta_3 \log \bar{y}_{c,t} \times G_{c,t} + \nu_c + \epsilon_{c,t} . \tag{4}$$

The logit transformation of the dependent variable $s_{c,t}$ maps the population share above the poverty line from the range 0–1 onto an unbounded range which can be conveniently fit with a linear model. (A broadly similar model, albeit without the logit transformation, was used by Campagnolo & Davide[34].)

We compile a data set of 1160 country-year observations of poverty rates, Gini coefficients[73,74] and GDP/capita values[75] from 131 countries and use it to fit the model with the above specification. Our model provides an excellent fit to the data (adjusted $R^2 = 0.93$) and shows that as expected both average income and Gini coefficient are highly significant drivers for poverty outcomes (see Section 8 of the SI for details). Note that there are many other variables that are potentially important drivers for poverty, for example economic structure, education and institutional quality. Any time-invariant differences between countries are already captured by our country fixed effects. Some of the time-varying effects, e.g. education and a number of policy variables, are already included as drivers for the SSP Gini coefficients[68], and as such they are also included indirectly in our poverty projections.

Note that we refrain from calculating the poverty shares directly from the cumulative distribution function of income for the following reasons. (i) It is likely that the tails of the distribution will deviate from the assumed lognormal form to a certain extent. As the bottom tail is particularly relevant for our analysis, this could lead to a bias. (ii) For obtaining reliable poverty estimates directly from the distribution, average incomes from household surveys would have to be used for fixing the mean of the distribution. As we are interested in future projections where household surveys are by construction unavailable, we index the mean of the distribution to GDP/capita, and work with GDP/capita projections from the SSPs.

**Projecting poverty headcounts and uncertainties.** We project national and global poverty headcounts in the baseline and mitigation scenarios and the respective differences as follows:

(1) The regression model is used to calculate the baseline projections for the share of the population in a given country above the poverty line, $s_{c,t}$ from the average income and Gini scenario data.

(2) National poverty headcounts are then given by

$$P_{c,t} = N_{c,t}(1 - s_{c,t}) , \tag{5}$$

where $N_{c,t}$ is the population. Regional and global poverty figures are the appropriate sums of national headcounts.

(3) Steps (1) and (2) are also repeated using the average income and Gini for the mitigation scenarios as calculated above, leading to the poverty projections for the mitigation scenario.

(4) The difference between poverty figures in mitigation and baseline scenario can be attributed to the effects of climate policy.

The regression model also provides us with an estimate of the uncertainty in the relationship between average income and Gini coefficient and poverty outcomes. Based on this we compute 68% prediction intervals (including approximately one standard deviation around the central estimate for future projections given the regression model) for the national, regional and global poverty figures (see Section 9 of SI for details). We also calculate uncertainties for the differences between the respective policy and baseline scenario. Due to the correlation between policy and baseline results the uncertainty of the difference between policy and baseline is smaller than either individual uncertainty. This allows us to compute the additional poverty caused by mitigation policies with fairly high precision despite the larger uncertainty on the policy and baseline projections.

**Reporting summary.** Further information on research design is available in the Nature Research Reporting Summary linked to this article.

## Data availability

The IAM scenario data analysed in this paper are computed with models that are available open source (see SI for links to code repositories). The data on energy and food expenditures by income group (see Section 5.2 of SI for details) were kindly provided by the authors of Dorband et al.[33], who in turn derived them from the World Bank's Global Consumption Database[69]. The data shown in the main figures of this study are available in this repository: https://doi.org/10.5281/zenodo.4320973. Intermediate data sets generated or analysed in this study are available from the corresponding author on reasonable request.

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

## Acknowledgements

The authors are grateful to Ira Dorband, Lorenzo Montrone and Jan Steckel for sharing the data compiled in Dorband et al.[33], and also for helpful support and valuable discussions concerning the further processing of these data. The authors also thank the

members of the REMIND and MAgPIE groups for helpful discussions and/or technical support, in particular Lavinia Baumstark, Christoph Bertram, David Klein, Hermann Lotze-Campen, Franziska Piontek, Miodrag Stevanovic and Isabelle Weindl. This work has been funded through the projects SHAPE, CHIPS and NAVIGATE. SHAPE and CHIPS are part of AXIS, an ERA-NET initiated by JPI Climate. SHAPE is funded by FORMAS (SE), FFG/BMWFW (AT), DLR/BMBF (DE, Grant No. 01LS1907A), NWO (NL) and RCN (NO) with co-funding by the European Union (Grant No. 776608). CHIPS is funded by FORMAS (SE), DLR/BMBF (DE, Grant No. 01LS1904A), AEI (ES) and ANR (FR) with co-funding by the European Union (Grant No. 776608). NAVI-GATE is funded by the European Union's Horizon 2020 research and innovation pro-gramme under grant agreement No. 821124.

## Author contributions

B.S. and E.K. designed the research with contributions from B.L.B., N.B., M.L. and A.P. B.S. performed the analysis. B.S. wrote the paper with contributions from all authors.

## Funding

## Competing interests

The authors declare no competing interests.
