## [Peer Review File · Nature Communications]

REVIEWER COMMENTS

Reviewer #1 (Remarks to the Author):

The paper analyses the interlinkages between SDG13 and SDG1. The authors find that meeting the 1.5° C target of Paris agreement will increase the number of people in absolute poverty. Recycling the carbon revenues equal-per-capita can compensate this side effect in 2030, but not in all regions; furthermore this compensative effect will fade out up to 2050. Using revenues from land use emission policies and international climate finance can be an effective way to achieve the double dividend of reducing emissions and poverty prevalence in 2030, but it will not again sufficient in 2050.

The quantitative assessment of the implication of mitigation targets on achieving poverty eradication is a valuable contribution to the IAM literature and can give interesting insights to policymakers dealing with SDG agenda. The paper is well written and communicate results in a straightforward way.

The methodology is convincing, combining the modelling exercise with empirical estimates. My concern is on how is motivated the section "Policy cost metrics". Four elements are considered affecting the household income and its distribution (GDP loss, price change of food and energy and carbon revenues). The problem is that income is related only to GDP loss that synthesise all price and quantity changes following the policy and carbon revenues; the price change effects do not have additional implications on income. It would be preferable to view the policy impact as a welfare change of households that consider both effects on income and on expenditure.

I suggest reading the paper Chen and Ravallion (2004) "Welfare Impacts of China's Accession to the World Trade Organization", The World Bank Economic Review, Vol. 18, No. 1. Despite the differences in the framework and research question, they decompose the implication of a policy on household welfare.

The used methodology could be applied to different policy questions, helping shading light on interactions and conflicts across SDGs and could offer a valuable support for policymaking.

To conclude, I would recommend the acceptance of the paper after comments and questions have been addressed.

Below are listed more specific questions and comments:

Line 119: the redistribution of revenue proportionally to income is not a really interesting scenario; it would be much more interesting redistribute revenues inversely proportional to income or only to the poorest deciles.

Line 169: the statement "if we assume that there is a revenue from carbon pricing..." is a bit unprecise; there is always a revenue from carbon pricing, the matter is more if the revenue is redistributed or not.

Line 380: there is a similar paper analysing the trade-off between poverty and climate targets and reaching similar conclusions:

Campagnolo L. and Davide M. (2019), Can Paris deal boost SDGs achievement? An assessment of climate mitigation co-benefits or side-effects on poverty and inequality, World Development Journal, Volume 122, pp 96-109

Line 641: equation 2 is difficult to justify according to the economic theory, it would be preferable to describe the policy impact on households as a welfare change.

Reviewer #2 (Remarks to the Author):

Summary and major comments

This paper uses simulation modeling to address the important issue of how climate mitigation

could affect poverty rates through its effects on income, food and energy prices. It shows that along the baseline socio-economic scenario (SSP2) there will be large decrease in poverty, measured relative to the current global poverty line of \$1.50/day. Climate mitigation would somewhat increase poverty relative to this baseline, unless the revenues generated by pricing carbon are redistributed to households, in which case the change in poverty due to climate mitigation can essentially be eliminated. It is important to highlight that the impacts of climate change itself on poverty are not included in this study.

I find the analysis is competently executed and am generally supportive of publication, given the importance of the question and that the methods used are fairly rigorous. If I am being hyper-critical, then I would observe that the particular model employed here is perhaps not ideally suited to addressing poverty questions, given the lack of spatial and commodity/sector resolution. The authors make the reasonable observation that other models like CGE models would be better suited to picking up some of these issues, but that such models tend not to cope with dynamics very well. Then again, this modeling exercise focuses mostly on a 2030 horizon, where arguably it's more important to have the resolution and the key mechanisms driving poverty, rather than necessarily the long-term dynamics. Another hyper-criticism is that one could read the results as saying climate mitigation is of second-order importance to poverty, compared with (i) socio-economic trends and (ii) redistributive policy both globally and nationally. Then again, we wouldn't have known this without this study, so fine.

The main concerns I developed while reading the main body of the paper are addressed towards the end of the paper, to be fair. I would, however, like to raise the following points, which I think the authors should consider:

- I worry about results that are based on assuming a constant poverty line of \$1.90/day. I assume the model works with real prices, therefore inflation is not an issue. I therefore think \$1.90 is reasonable on a 2030 timescale, but not on a 2050 timescale. I suggest either dropping the 2050 horizon, which isn't the main focus anyway, or building in a rate of increase from 2030 onwards.
- I worry even more about the messaging around recycling carbon price revenues. My concern is that readers who are not expert in public finance will form the view that the only way to make carbon pricing compatible with alleviating poverty is through specific ear-marking or hypothecation of tax revenues for e.g. a per-capita carbon dividend. Such ideas have become popular as a way of overcoming public opposition to carbon pricing, but from the point of view of public finance they are a bit of a nightmare, because they diminish the ability of the finance ministry to optimally allocate revenue between e.g. schools, health, defense, R&D, etc. The point is that the baseline scenario seems to implicitly assume the revenues are somehow swallowed by the government and don't resurface in any form, whereas any government that tries to structure public spending/taxation in a progressive way through e.g. the income tax schedule or spending on basic public services and social safety nets can alleviate poverty this way instead. I think the authors need to frame their results so that this comes across. The baseline is unrealistic.
- I think the problem about the extension to pricing agricultural emissions is that this really stretches the credibility of some of the core assumptions. As well as those mentioned in the paper, I am thinking of the assumption of $\alpha=1$ – distributionally neutral GDP losses due to climate mitigation – in the context of poverty in Sub Saharan Africa. In SSA, most households below the poverty line will be agricultural and thus wouldn't such a policy have a material effect on their incomes relative to other households?
- Given the baseline socio-economic trend seems to be more important for poverty alleviation than climate policy, it may add value to look at alternative SSPs (while recognizing that not all SSPs are consistent with 1.5C). A similar comment applies to possibly looking at alternative burden-sharing regimes between rich and poor countries in terms of differential carbon prices.
- Is there some double-counting of energy and food expenditures when there is also a separate term capturing the overall impact on GDP? Assuming no borrowing/savings response, the GDP impact should correspond with the aggregate impact of spending on different commodities, thus price x quantity, where two commodities in the basket are food and energy, both large

components presumably for poor households.

Other comments

- The claim on line 55 that this paper is the first to “robustly quantify the consequences of a Paris-compatible mitigation pathway for global poverty” is over-selling the paper somewhat, given the limitations of the modelling. I suggest the authors find a more targeted statement of their contribution.
- My understanding is that Equation (1) applies to each of the four deltas? If so, to avoid confusion I suggest you index the delta in Equation (1) with something like i or j , where i is GDP, FE, food, GHG.
- Where “additional XX million” is written this should always be prefaced with “an”.

Reviewer #3 (Remarks to the Author):

The manuscript simulates the evolution of poverty as absolute poverty in an IAM comparing the baseline projection and different climate policies. Based on a parametric distribution within countries and an empirical link between inequality, GDP growth and poverty, the authors find that climate policies could lead to an increase in absolute poverty by 50 million people in 2030 while using the carbon price revenues for redistribution could counterbalance this increase or even slightly lead to a net decrease.

Overall, the topic is certainly very relevant, and the paper the first attempt to simulate poverty in a climate IAM framework. Moreover, the paper is very well written. However, I have to major concerns with the methodology of the paper:

- 1) The way poverty is projected is however quite simple and does not seem to contain any structural (say, economic structure, education, gender equality, alphabetization, etc.) nor policy (institutions, tax and transfer systems, ODA, cash transfer schemes, ...) drivers of poverty in its simulation, which are both of course highly relevant for poverty.
- 2) The treatment of uncertainty is suggesting very narrow confidence bands, but includes only very minor drivers of uncertainty, namely the cross-section regression and the country-regional aggregation procedure. Poverty in the future will depend highly on many factors, and even considering only inequality, Rao et al. (2018) have shown that inequality could have very different trajectories even in the near term. Hence, I feel a bit uncomfortable with the presented results suggesting (a) a strong decline in poverty and (b) little variation across the estimates of climate policy impacts. Notably, at least considering for instance different SSPs (added to the main figure/results from the SI) or comparing the results with some of the few available CGE based projections of poverty in the baseline scenarios would be highly valuable in order to interpret and gauge the results of the paper in the context of the development literature and scenario ensembles.
- 3) The naming and interpretation of the recycling schemes: If I am not mistaken, typically a lump sum is the mostly neutral standard (which you call equal per capita), which should not affect effort etc. In your case it becomes the most redistributive scheme, and I see the point, just to note. The “proportional to income” is quite nonsensical given no such scheme exists. You mention it could reflect a VAT reduction, but so I would suggest “consumption based” or so, but just a suggestion. In development economics income is hardly measurable and if anything used to redistribute inversely proportional to income (for instance, in cash transfer or proxy based means testing transfer schemes). In general, such transfers could of course lead to huge poverty co-benefits that are themselves unrealistic (even though could be interesting to see its potential!), but the wording is a bit confusing here, but just a suggestion. Finally, “no recycling” is also a bit problematic, so you just “destroy” the money and it doesn’t show up at all in the income of people? Even with transfers say only to the elite or so somehow the money needs to be in the economy, maybe considering it giving it to the top 5 per cent or so (which of course doesn’t change poverty rates) could make at least sense to avoid it is just “lost”.

I do think however that both issues could be addressed by explained the model a bit better in the Methods part and possibly adding some further uncertainty analysis, in which case I can see the

paper well suited for publication in Nature Communications.

Further comments:

- In the abstract, "residual increase" is not clear, and "these" measures is not clear what "these" encompasses.
- In figure 1, India seems to have less than 5% or so absolute poor, while the last available data indicates 21%. This would be a huge success and should at least be discussed or checked if makes sense.
- The carbon pricing you implemented is not clear to me. Is it a tax or permit scheme? You mention the differentiation but How this works is important for the results and should be shortly explained in the main paper as well.
- You mention several times that carbon revenues decline. However, I assume your carbon price follows some Hotelling like rule so not sure what happens to the revenues, plus of course there are negative emissions, and the carbon neutrality is (I assume) implemented only globally. Some figure on the carbon price or total revenues could be helpful.
- Line 216: please explain "cost-only counterfactual" or rename.
- Line 242: What happens in China?
- Figure 3: "68% prediction intervals" should be explained.
- In Figure 4 (and subsequent), the y axis would be better labelled "Additional people in poverty [million]"
- Line 258: explain "strong differentiation in carbon prices" as it seems highly driving the regional results.
- Line 299: What you mean by "Combined with a climate dividend"?

On the distributional modelling part:

Line 623, The four δ ... Values seems something is missing there.

Equation (1) should be better explained. Not sure about the delta there, and the exponential term looks like some kind of lognormal integration, but it should be mentioned. Also units of observations would be good, or as suffix or in the text, I assume these are regions, but then what is sigma and mju?

Line 636: should read "income" instead of probability distribution

Equation 3, also only what unit of observation are reported, individual or averages?

Equation 4, just the reason for the LHS specification should be briefly mentioned in the text.

In line 702, I would add " across model regions" to make clear why the Monte carlo at country level and how it aggregates.

Manuscript resubmission “Combining ambitious climate policies with efforts to eradicate poverty” (Soergel et al., 2020)

Response to editor and reviewer comments

We would like to thank the three reviewers for their helpful and constructive feedback. We believe that addressing their comments has strengthened our paper and improved the clarity of the presentation of our results.

We have made the following main changes:

- We clarified the presentation of our distributional framework, and adapted the choice of revenue recycling scenarios displayed in the results.
- We moved our discussion of alternative SSPs from the supplement to the main text and extended it substantially.
- We merged our discussion of a higher poverty line (previously in the supplement) with our discussion of longer-term poverty trends.

Note that these changes involved some rearranging of content in the results section to maintain a coherent presentation of our results. Thus larger text blocks are marked as changed in the revised version (green colour highlighting), although much of the underlying content did not change substantially.

A detailed explanation of our changes is given in the point-by-point response to the reviewers below.

[...]

Reviewer #1 (Remarks to the Author):

The paper analyses the interlinkages between SDG13 and SDG1. The authors find that meeting the 1.5° C target of Paris agreement will increase the number of people in absolute poverty. Recycling the carbon revenues equal-per-capita can compensate this side effect in 2030, but not in all regions; furthermore this compensative effect will fade out up to 2050. Using revenues from land use emission policies and international climate finance can be an effective way to achieve the double dividend of reducing emissions and poverty prevalence in 2030, but it will not again sufficient in 2050.

The quantitative assessment of the implication of mitigation targets on achieving poverty eradication is a valuable contribution to the IAM literature and can give interesting insights to policymakers dealing with SDG agenda. The paper is well written and communicate results in a straightforward way.

The methodology is convincing, combining the modelling exercise with empirical estimates.

We thank the reviewer for the positive and constructive feedback. We will address their individual points in detail below.

My concern is on how is motivated the section “Policy cost metrics”. Four elements are considered affecting the household income and its distribution (GDP loss, price change of food and energy and carbon revenues). The problem is that income is related only to GDP loss that synthesise all price and quantity changes following the policy and carbon revenues; the price change effects do not have additional implications on income. It would be preferable to view the policy impact as a welfare change of households that consider both effects on income and on expenditure.

I suggest reading the paper Chen and Ravallion (2004) “Welfare Impacts of China's Accession to the World Trade Organization”, The World Bank Economic Review, Vol. 18, No. 1. Despite the differences in the framework and research question, they decompose the implication of a policy on household welfare.

This is a helpful suggestion on how to improve the clarity of the presentation of our distributional model, as indeed the core idea of our model resembles the approach by Chen & Ravallion (2004). We use the term ‘real income’ to indicate that our metric includes changes both on the income and the expenditure side (see also Eq.1 and associated discussion in Hussein et al. 2013). Changes in real income occur through (1) changes in income, (2) changes in energy and food prices, (3) payments received from the recycling of the carbon pricing revenue (extending the approach from Leimbach et al. 2018). We improved our explanation of this approach, which now includes that this can also be described equivalently as a welfare change as in Chen & Ravallion (see the revised sections “Policy cost metrics” and “Distribution of policy costs and revenues” in Methods).

Some further notes on this point:

- While Chen & Ravallion use a welfare framework to derive their approach, ultimately they use income as a monetary metric of utility to analyse the poverty and inequality implications (similar to our approach).

- The welfare change of Chen & Ravallion (their Eq. 3) is derived under the assumption of small price changes, therefore neglecting welfare impacts through price-induced quantity changes. This approximation does not apply in our case, where price changes for energy commodities are large (typically +50-100%). Thus also quantity changes matter, and we take them into account.
- Our global long-term modelling framework REMIND-MAGPIE does not provide the high sectoral resolution of a CGE model, and detailed household data is of course unavailable for our analysis. We therefore use the reduction in GDP/capita as a proxy for average income loss. A more detailed coverage of the income side, especially for agricultural incomes, and of the employment dynamics, would certainly be a desirable extension for future work.

The used methodology could be applied to different policy questions, helping shading light on interactions and conflicts across SDGs and could offer a valuable support for policymaking.

To conclude, I would recommend the acceptance of the paper after comments and questions have been addressed.

Below are listed more specific questions and comments:

Line 119: the redistribution of revenue proportionally to income is not a really interesting scenario; it would be much more interesting redistribute revenues inversely proportional to income or only to the poorest deciles.

The redistribution “proportional to income” represents a case of climate policy without associated progressive redistribution. To clarify that it is distributionally neutral (not changing the level of inequality), we have renamed it to “neutral” throughout the text and in all figures.

Following the suggestion, we have introduced a new scheme redistributing inversely proportional to income (“strongly progressive”)’ see Sections “Options for generating poverty co-benefits”/”More progressive redistribution” and the revised Fig. 5.

Finally, addressing the suggestions of Reviewers 2 and 3, we have removed the “no recycling” scheme from our results, to avoid the misunderstanding that it represents a baseline scenario where the carbon pricing revenues are “lost”.

Line 169: the statement “if we assume that there is a revenue from carbon pricing...” is a bit unprecise; there is always a revenue from carbon pricing, the matter is more if the revenue is redistributed or not.

In revising our naming and description of the revenue recycling scenarios (previous point), this point has been addressed as well (see beginning of second paragraph of section “Global poverty headcount”)

Line 380: there is a similar paper analysing the trade-off between poverty and climate targets and reaching similar conclusions:

Campagnolo L. and Davide M. (2019), Can Paris deal boost SDGs achievement? An assessment of climate mitigation co-benefits or side-effects on poverty and inequality, World Development Journal, Volume 122, pp 96-109

This is a very helpful reference. We now mention this paper in the introduction and in the discussion, and have included references at the appropriate places. We note that our work differs from Campagnolo & Davide (C&D) in the following main points:

- scope: C&D investigate the poverty side-effects of countries' conditional NDCs, which are, however, by no means ambitious enough to reach the Paris target (and thus also inconsistent with the SDG agenda). Our analysis looks at the 1.5°C target, which requires a much faster decarbonization, with the potential for substantially higher poverty side-effects.
- full distributional analysis: Our analysis captures the direct distributional effects of climate policy on poverty by modelling the effects of energy & food prices and revenue recycling on the distribution of income. As we understand the C&D paper, they do not actually perform such a direct distributional analysis. Instead they assess the effects of climate policy on inequality through a regression model between the Palma ratio and CGE model outputs like public education expenditure, agricultural value added, etc. It seems that the potentially strongly progressive effect of the revenue recycling is thus not fully captured in the C&D analysis. Our analysis demonstrates that a progressive use of carbon pricing revenues is key to compensate poverty side-effects.
- near-term poverty projections: C&D project a global poverty headcount of 80 million people in 2030 in their SSP2-baseline scenario. Given recent historical trends (already prior to the COVID-19 pandemic), this seems very low. For comparison, our model projects 350 million for the SSP2 baseline in 2030. Crespo Cuaresma et al. (2018), based on a different methodology and assumptions, project even higher numbers.

Line 641: equation 2 is difficult to justify according to the economic theory, it would be preferable to describe the policy impact on households as a welfare change.

We substantially extended our explanation and discussion of the distribution of policy costs, including the mapping to the welfare change of Chen & Ravallion (see detailed response above). Specifically for Eq. 2, note that our changes in FE/food expenditures include the carbon prices (now also clarified in the text). In the simplifying example of a single representative household, and single commodities in FE and food sector with perfectly inelastic demands, the Δ_{FE} , Δ_{food} and Δ_{GHG} terms would exactly cancel, such that only the Δ_{GDP} term remains on the RHS (see also the discussion at the end of Section 6 of the supplement).

Reviewer #2 (Remarks to the Author):

Summary and major comments

This paper uses simulation modeling to address the important issue of how climate mitigation could affect poverty rates through its effects on income, food and energy prices. It shows that along the baseline socio-economic scenario (SSP2) there will be large decrease in poverty, measured relative to the current global poverty line of \$1.50/day. Climate mitigation would somewhat increase poverty relative to this baseline, unless the revenues generated by pricing carbon are redistributed to households, in which case the change in poverty due to climate mitigation can essentially be eliminated. It is important to highlight that the impacts of climate change itself on poverty are not included in this study.

I find the analysis is competently executed and am generally supportive of publication, given the importance of the question and that the methods used are fairly rigorous. If I am being hyper-critical, then I would observe that the particular model employed here is perhaps not ideally suited to addressing poverty questions, given the lack of spatial and commodity/sector resolution. The authors make the reasonable observation that other models like CGE models would be better suited to picking up some of these issues, but that such models tend not to cope with dynamics very well. Then again, this modeling exercise focuses mostly on a 2030 horizon, where arguably it's more important to have the resolution and the key mechanisms driving poverty, rather than necessarily the long-term dynamics. Another hyper-criticism is that one could read the results as saying climate mitigation is of second-order importance to poverty, compared with (i) socio-economic trends and (ii) redistributive policy both globally and nationally. Then again, we wouldn't have known this without this study, so fine.

We thank the reviewer for their positive feedback and helpful comments on our paper. Indeed socio-economic trends are the most important driver for poverty. Therefore we have now included different SSP scenarios into our analysis (see also the detailed response on this point below). However, while mitigation policies are not the prime driver for poverty trends themselves, their effects on poverty are of prime importance for the question how to mitigate climate change in accordance with the broader sustainable development agenda.

The main concerns I developed while reading the main body of the paper are addressed towards the end of the paper, to be fair. I would, however, like to raise the following points, which I think the authors should consider:

- I worry about results that are based on assuming a constant poverty line of \$1.90/day. I assume the model works with real prices, therefore inflation is not an issue. I therefore think \$1.90 is reasonable on a 2030 timescale, but not on a 2050 timescale. I suggest either dropping the 2050 horizon, which isn't the main focus anyway, or building in a rate of increase from 2030 onwards.*

Indeed the model works with real prices, but we agree that a higher poverty line for 2050 would be a valuable addition to our analysis. We have implemented the following changes:

- We removed the 2050 results from Fig. 4 and the associated discussion of our results in the sections “Global poverty headcount” and “Regional poverty trends”. Instead, we now discuss the variation across SSP dimensions there (see also below).
- For Fig. 5 and associated discussion (revised section “Options for generating poverty co-benefits”) we kept the 2050 results, as we believe there is valuable additional information in how the availability of carbon pricing revenues determines the ability to compensate for policy side effects. Here we stuck to the \$1.90 line to allow for direct comparison of the numbers between 2030 and 2050.
- We added a new section “Higher poverty line and longer-term prospects for poverty eradication”. Here we use a poverty line of \$5.50/day, which follows the current World Bank poverty threshold for current upper-middle income countries. (Note that we opted for a fixed higher threshold instead of a dynamic one, as the latter would make time series graphs difficult to interpret.) The new Fig. 6 summarizes our results for a higher poverty line (previously shown in the SI) and focuses on the 2050 horizon.

• I worry even more about the messaging around recycling carbon price revenues. My concern is that readers who are not expert in public finance will form the view that the only way to make carbon pricing compatible with alleviating poverty is through specific ear-marking or hypothecation of tax revenues for e.g. a per-capita carbon dividend. Such ideas have become popular as a way of overcoming public opposition to carbon pricing, but from the point of view of public finance they are a bit of a nightmare, because they diminish the ability of the finance ministry to optimally allocate revenue between e.g. schools, health, defense, R&D, etc. The point is that the baseline scenario seems to implicitly assume the revenues are somehow swallowed by the government and don't resurface in any form, whereas any government that tries to structure public spending/taxation in a progressive way through e.g. the income tax schedule or spending on basic public services and social safety nets can alleviate poverty this way instead. I think the authors need to frame their results so that this comes across. The baseline is unrealistic.

Concerning the ‘no recycling’ scenario in the original submission: indeed it assumed that the revenues are not recycled in any form. It was, however, not intended as a baseline, but rather as a counterfactual showing only the “cost” side of mitigation policies (allowing for a decomposition of the net effect into the effects of costs + revenue recycling). But we do see the point that this is easily misunderstood, and have thus decided to remove this scenario from the results. We have also renamed the other revenue recycling scenarios (“proportional to income” -> “neutral”, “equal per capita” -> “progressive”) to better reflect their effect on the income distribution.

Concerning the public finance perspective: We have added the following paragraph on this to our discussion: “Instead of directly redistributing the carbon pricing revenues, governments could also use them to increase their spending for other poverty-reducing policies. This includes for example education spending, but also infrastructure development that is critical for achieving other SDGs, such as access to electricity, clean water, sanitation, transport and telecommunication (Franks et al. 2018, Dorband et al. 2017, Thacker et al. 2019). While we do not attempt to directly quantify the effect of such policies, our different redistribution schemes can be seen as stylized explorations of different degrees of progressivity in spending the revenues from carbon pricing.”

Finally, we note that our SSP baselines indirectly include the degree of progressivity of public spending through the SSP Gini scenarios by Rao et al. These scenarios contain policy variables such as public spending for education, health and social services as drivers (see also changes to Methods section “Regression model for poverty outcomes” and response to Reviewer 3 below). As such, varying the SSP scenarios captures both the effects of faster/slower national income growth and (indirectly) the government decisions determining how it is distributed.

• I think the problem about the extension to pricing agricultural emissions is that this really stretches the credibility of some of the core assumptions. As well as those mentioned in the paper, I am thinking of the assumption of $\alpha=1$ – distributionally neutral GDP losses due to climate mitigation – in the context of poverty in Sub Saharan Africa. In SSA, most households below the poverty line will be agricultural and thus wouldn't such a policy have a material effect on their incomes relative to other households?

This raises an important point that we already touched upon briefly in the discussion section. GHG-equivalent pricing of agricultural emissions, and land-based mitigation measures (bioenergy, afforestation) will increase the prices for agricultural goods, and could thus in principle also lead to higher revenues for agricultural households (see also references in discussion section). In this case, the assumption of distributionally neutral GDP losses would indeed not be justified. However, we argue that this is not a substantial effect for poor agricultural households: The GHG pricing revenues do not increase their income, and the increased demand for land mainly benefits larger land-owners (see e.g. the discussion in Hussein et al. 2013). Another important distinction in this context is the one between urban and rural poor, whose income and expenditures might be affected differently by mitigation policies. Unfortunately, no separate data for urban/rural poverty was available for the international poverty line used in our analysis. We do, however, fully agree with the general point that a more detailed quantification on the income side would be a desirable extension of the work presented in our paper. If such a quantification should become available, it can be readily incorporated into our framework by specifying the appropriate elasticity. (We added this in the Discussion section.)

- *Given the baseline socio-economic trend seems to be more important for poverty alleviation than climate policy, it may add value to look at alternative SSPs (while recognizing that not all SSPs are consistent with 1.5C). A similar comment applies to possibly looking at alternative burden-sharing regimes between rich and poor countries in terms of differential carbon prices.*

We have included different SSPs into our analysis, which now covers all SSPs for the baseline scenarios without climate policies (see revised Section “Poverty trends in reference scenarios” and revised Fig. 1). We furthermore show SSP1, SSP2 and SSP5 for the effect of mitigation policies (see revised Sections “Effects of climate policy and redistribution” and following, and revised Fig. 4). These three SSPs span the axis of mitigation challenges in the SSP matrix. (At the time of writing, no parametrization of SSP3 and SSP4 is available for REMIND-MAgPIE.) We removed the discussion of alternative SSPs from the SI to avoid duplication with the main manuscript.

Concerning the burden sharing, we already model the effects of a climate finance scheme with international transfers on top of our price differentiation. We also refer to N. Bauer et al., *Quantification of an efficiency-sovereignty trade-off in climate policy* (Nature, in review) for a further discussion of carbon price differentiation and burden sharing.

- *Is there some double-counting of energy and food expenditures when there is also a separate term capturing the overall impact on GDP? Assuming no borrowing/savings response, the GDP impact should correspond with the aggregate impact of spending on different commodities, thus price x quantity, where two commodities in the basket are food and energy, both large components presumably for poor households.*

Our approach decomposes the total change in real income (or equivalently, welfare measured in a monetary metric) into changes on the income and expenditure side. Given that our IAM, unlike a CGE model, does not have a high sectoral resolution of different production sectors, we use the overall GDP loss to measure the effect of climate policy on the income side. The food and energy terms, on the other hand, capture the expenditure side (note that the FE and food expenditures include the carbon prices paid, so there is an extra term for the redistribution of the revenues). All of these terms are relevant for capturing the distributional impact of the policies, so we do not think there is a double counting here. (See revised Methods sections “Policy costs metrics” and “Distribution of policy costs and revenues”, and the more detailed discussion in response to Reviewer 1).

Other comments

- *The claim on line 55 that this paper is the first to “robustly quantify the consequences of a Paris-compatible mitigation pathway for global poverty” is over-selling the paper somewhat, given the limitations of the modelling. I suggest the authors find a more targeted statement of their contribution.*

We changed this to “our study quantifies the consequences of an ambitious, Paris-compatible mitigation pathway for global poverty until mid-century”.

- *My understanding is that Equation (1) applies to each of the four deltas? If so, to avoid confusion I suggest you index the delta in Equation (1) with something like i or j , where i is GDP, FE, food, GHG.*

That’s correct, it applies to each of the deltas. We added an index j .

- *Where “additional XX million” is written this should always be prefaced with “an”.*

Done.

Reviewer #3 (Remarks to the Author):

The manuscript simulates the evolution of poverty as absolute poverty in an IAM comparing the baseline projection and different climate policies. Based on a parametric distribution within countries and an empirical link between inequality, GDP growth and poverty, the authors find that climate policies could lead to an increase in absolute poverty by 50 million people in 2030 while using the carbon price revenues for redistribution could counterbalance this increase or even slightly lead to a net decrease. Overall, the topic is certainly very relevant, and the paper the first attempt to simulate poverty in a climate IAM framework. Moreover, the paper is very well written. However, I have to major concerns with the methodology of the paper:

We thank the reviewer for their positive and helpful feedback, and will respond to the individual points in detail below.

- 1) *The way poverty is projected is however quite simple and does not seem to contain any structural (say, economic structure, education, gender equality, alphabetization, etc.) nor policy (institutions, tax and transfer systems, ODA, cash transfer schemes, ...) drivers of poverty in its simulation, which are both of course highly relevant for poverty.*

We fully agree that the mentioned structural and policy variables are potentially important drivers for poverty. We did not include them explicitly in our analysis for a number of reasons. (We have made points a) and b) more explicit in our description of the model, see Methods section “Regression model for poverty outcomes”)

- (a) Country fixed effects: Time-independent structural differences across countries are already captured in the poverty regression model as country fixed effects. Time-dependent structural effects could be captured by recognising and extrapolating these differences using a parameterized

approach. Here we do not represent these structural characteristics explicitly, but implicitly through the SSP Gini scenarios (see point b).

- (b) Gini projections: The Rao et al. Gini projections used in our analysis already cover a number of these structural variables (education, policy) as drivers for future inequality. Therefore, also a variation in these drivers is implicitly already considered in our projections through the Gini coefficient.
- (c) Scenario data availability: Of the mentioned variables, only education is covered in the SSPs and thus available for our projections. We explicitly tested including the primary education completion rate as an additional predictor in our regression model, but found that its coefficient is not significant, and that its inclusion doesn't increase the goodness of fit substantially. (Note that education completion rates, and likely some of the other variables as well, have a significant correlation with GDP/capita and are therefore to a large extent implicit to our estimates. Their explicit inclusion in the regression model would on the other hand open up problems of multicollinearity). Furthermore, the country-years with available education data differ from those with inequality/poverty data, which would further reduce the size of our data set - in particular to only a single observation for a number of important African and Asian countries. Therefore including education data would not be an improvement to our analysis.
- (d) Effect of climate policy: How these variables would change with mitigation policies, and thus their importance for our analysis, is not yet robustly quantified. While climate policies will influence some of the mentioned drivers (e.g. economic structure), here we focused on those channels where a robust quantification is already possible (i.e. average income, inequality).

2) *The treatment of uncertainty is suggesting very narrow confidence bands, but includes only very minor drivers of uncertainty, namely the cross-section regression and the country-regional aggregation procedure. Poverty in the future will depend highly on many factors, and even considering only inequality, Rao et al. (2018) have shown that inequality could have very different trajectories even in the near term. Hence, I feel a bit uncomfortable with the presented results suggesting (a) a strong decline in poverty and (b) little variation across the estimates of climate policy impacts. Notably, at least considering for instance different SSPs (added to the main figure/results from the SI) or comparing the results with some of the few available CGE based projections of poverty in the baseline scenarios would be highly valuable in order to interpret and gauge the results of the paper in the context of the development literature and scenario ensembles.*

Indeed our confidence bands show only the uncertainty related to our projection exercise, but not the “uncertainty” of future socio-economic developments as captured by the SSPs. Arguably, the latter is a different “type” of uncertainty, and cannot be captured in a probabilistic framework.

Nonetheless we agree that discussion of the different SSPs will enhance the value of our analysis. We now show all SSPs for the baseline trends (see revised Fig. 1 and section “Poverty trends in reference scenarios”), and SSP1, SSP2 and SSP5 for the

effect of mitigation policies (see revised Fig.4 and sections “Effect of climate policy and redistribution” and following). These three SSPs span the axis of mitigation challenges in the SSP matrix. (At the time of writing, no parametrization of SSP3 and SSP4 is available for REMIND-MAgPIE.) We removed the discussion of alternative SSPs from the SI to avoid duplication with the main manuscript.

We also include a sentence into the caption of Fig. 1b (first time uncertainty bands are shown) that describes which uncertainty is represented by these bands, and refer to this again in the caption of Fig. 3.

3) *The naming and interpretation of the recycling schemes: If I am not mistaken, typically a lump sum is the mostly neutral standard (which you call equal per capita), which should not affect effort etc. In your case it becomes the most redistributive scheme, and I see the point, just to note. The “proportional to income” is quite nonsensical given no such scheme exists. You mention it could reflect a VAT reduction, but so I would suggest “consumption based” or so, but just a suggestion. IN development economics income is hardly measurable and if anything used to redistribute inversely proportional to income (for instance, in cash transfer or proxy based means testing transfer schemes). In general, such transfers could of course lead to huge poverty co-benefits that are themselves unrealistic (even though could be interesting to see its potential!), but the wording is a bit confusing here, but just a suggestion. Finally, “no recycling” is also a bit problematic, so you just “destroy” the money and it doesn’t show up at all in the income of people? Even with transfers say only to the elite or so somehow the money needs to be in the economy, maybe considering it giving it to the top 5 per cent or so (which of course doesn’t change poverty rates) could make at least sense to avoid it is just “lost”.*

We agree that the selection and naming of the redistribution schemes might have been confusing. We have thus implemented the following changes:

- We have removed the ‘no recycling’ scheme from our results. It was intended as a counterfactual allowing for a decomposition of the net effect into a “cost” and “revenue” side. But we agree that this is easily misunderstood, and we have thus decided to remove it for increased clarity in the presentation of our results (see also response to Reviewer 2, who had raised a similar point).
- The “proportional to income” scheme represents climate policy without associated progressive redistribution measures. We renamed it to “neutral”, reflecting that this scheme uses the revenues in a distributionally neutral way (not changing the degree of inequality in the distribution of income).
- We renamed “equal per capita” to “progressive” to clarify that it reduces the degree of inequality in the distribution of income.
- We added a “strongly progressive” scheme with redistribution inversely proportional to income; see the new Fig. 5, and the revised section “Options for generating poverty co-benefits”. (Again see also response to Reviewer 2.)

I do think however that both issues could be addressed by explained the model a bit better in the Methods part and possibly adding some further uncertainty analysis, in which case I can see the paper well suited for publication in Nature Communications.

As detailed above, we extended the description of the poverty regression model, added additional SSPs to represent the uncertainty in future socioeconomic development, and revised the selection and naming of the revenue recycling schemes.

Further comments:

- *In the abstract, “residual increase” is not clear, and “these” measures is not clear what “these” encompasses.*

We rewrote the last two sentences of the abstract for increased clarity.

- *In figure 1, India seems to have less than 5% or so absolute poor, while the last available data indicates 21%. This would be a huge success and should at least be discussed or checked if makes sense.*

Unfortunately there is no recent data from India - WDI reports 21% for 2011. In our scenario, GDP/capita more than doubles in IND in the period 2011-2030. Also, in this income range, poverty rates are still very sensitive to increases in income (flattening off at higher income levels). Comparing to the historical data for China: over a comparable GDP/cap range (years 2002-2012), poverty rates went from 32% to 6.5%, and to below 1% in 2015. In that light, our projection for India seems quite reasonable. The trend of rapidly falling poverty rates in India is also in accordance with the 2018 World Bank Poverty Report (their Fig. 1.9). We have added this last point in the text.

Motivated by this comment we have also examined the projection for Zimbabwe, which also displayed a rapid reduction in poverty. Unlike India, here the projection was indeed not robust, as there was only a single observation available in the historical period used to fit the poverty model. For this reason, we have removed Zimbabwe, and a small number of other countries with the same issue, from our sample. (No country contributing substantially to the global or regional poverty headcounts was among them, such that global and regional poverty headcounts remain essentially unchanged.)

- *The carbon pricing you implemented is not clear to me. Is it a tax or permit scheme? You mention the differentiation but How this works is important for the results and should be shortly explained in the main paper as well.*

It is a regionally differentiated carbon tax, whose level is adjusted endogenously such that the specified CO₂ budget is met (we clarified this in the Methods section “Scenario description”). Also, the differentiation is now shortly described in the main text as well (beginning of section “Effects of climate policy and redistribution”), and refers to the more detailed explanation in the Methods.

- *You mention several times that carbon revenues decline. However, I assume your carbon price follows some Hotelling like rule so not sure what happens to the revenues, plus of course there are negative emissions, and the carbon neutrality is (I assume) implemented only globally. Some figure on the carbon price or total revenues could be helpful.*

Our carbon price does in fact not follow a Hotelling path, mainly because the simple Hotelling picture is not appropriate anymore if negative emission technologies are available. Instead, we implement a peak budget approach (see description and references in Methods). Compared to an exponential Hotelling path, this leads to higher carbon prices in the near term, but lower prices in the longer term. A figure of the carbon price is already available as Suppl. Fig. 2 (we now referenced it more clearly).

In most of our model regions, carbon pricing revenues from the energy system decrease relative to GDP from ~2040 onwards, and approach zero towards ~2050 when net zero emissions is approached (see Suppl. Fig. 3 - now also referenced at the appropriate place in the main text).

- *Line 216: please explain “cost-only counterfactual” or rename.*

We removed this revenue recycling scheme (see detailed explanation above).

- *Line 242: What happens in China?*

China is a separate model region (now also clarified in the text). Also, as noted above, poverty rates in China are already below 1%, and thus it does not contribute to the global poverty headcount substantially any more.

- *Figure 3: “68% prediction intervals” should be explained.*

We now already show uncertainty bands in our revised Fig. 1, and therefore added an explanation of the uncertainty bands, and a reference to the corresponding Methods section (Projecting poverty headcounts and uncertainties) to its caption. The captions of Figs. 3 and 4 now refer to this information as well.

- *In Figure 4 (and subsequent), the y axis would be better labelled “Additional people in poverty [million]”.*

Yes, that’s clearer, thanks.

- *Line 258: explain “strong differentiation in carbon prices” as it seems highly driving the regional results.*

We added a brief explanation of the carbon pricing differentiation to the beginning of Section “Effects of climate policy and redistribution” (see also above).

[Note that we have restructured the section this comment originally referred to in order to allow for a more coherent presentation of the results added during the review.]

- *Line 299: What you mean by “Combined with a climate dividend”?*

The combination of international transfers and equal-per-capita redistribution (clarified in the text).

On the distributional modelling part:

Line 623, The four δ Values seems something is missing there.

We added an index j for increased clarity.

Equation (1) should be better explained. Not sure about the delta there, and the exponential term looks like some kind of lognormal integration, but it should be mentioned. Also units of observations would be good, or as suffix or in the text, I assume these are regions, but then what is sigma and μ ?

We added some explanation around Eq. 1, and refer to the detailed derivation in the SI more clearly. (Indeed it is based on a lognormal integration.) We also clarified in the text that the country-level distribution is modelled as lognormal, such that σ and μ are readily determined from average income and Gini coefficient. Note that Eq. 1 refers to a single person with baseline income y (think of y as a continuous random variable) - but we felt adding an extra index i here would just add notational clutter.

Line 636: should read “income” instead of probability distribution

Changed to “income distribution”.

Equation 3, also only what unit of observation are reported, individual or averages?

This refers to averages within one of four consumption groups in a given country (added this info in the text).

Equation 4, just the reason for the LHS specification should be briefly mentioned in the text.

We added a short explanation.

In line 702, I would add “across model regions” to make clear why the Monte carlo at country level and how it aggregates.

We do not aggregate the Monte Carlo simulations from country level to model regions, but instead use them to calculate national poverty projections (as shown in

Fig. 2). These then sum up to the regional and global numbers. (We added a clarification in the text.)

Other minor changes (in addition to changes following reviewer comments):

- We added a note to the discussion section that the impact of the COVID-19 pandemic is not yet included in our poverty numbers.
- We included recent data updates in the World Development Indicators into our data sets used for fitting (a) the estimate of the income elasticities, and (b) the poverty regression model. These small changes in the underlying data sets caused minimal changes in the resulting fit coefficients (now updated in SI), but without substantial effects on our reported results.
- We included the REMIND-MAgPIE model region containing the non-EU European countries into the list of OECD regions used to calculate the international climate finance transfers. This improves the mapping between model regions and OECD member countries. The effect on our results with international climate finance transfers is minimal.
- We harmonized the SSP Gini coefficients to the SSP2 values until 2020 to avoid a divergence of the scenarios in the historical period (for details see addition to Methods section “General distributional framework”). This leads to minor changes in the SSP1 and SSP5 results (previously in the SI, now in main text); our main SSP2 results remain unchanged.

REVIEWER COMMENTS

Reviewer #1 (Remarks to the Author):

The paper, in the revised version, looks extremely improved. Expanding the analysis to consider all SSPs is a value added for the paper and it allows understanding the range of policy outcomes under different socioeconomic development and mitigation efforts in place. Furthermore, the rephrasing of recycling scheme description improves greatly the understanding and the additional "strongly progressive" scheme conveys a strong message to that part of scholars which sees mitigation as necessarily detrimental to poverty reduction.

I am still not completely convinced by the rewording of "household income" into "real income". In Hussein et al. 2013, the use of this term seems more in line with its meaning, i.e. income cleaned by inflationary effects. But, in the paper, the Authors clearly state that they account for both price and quantity change of the consumed commodities. So I still think that "real income" is improperly used here.

Reviewer #2 (Remarks to the Author):

The authors have done a good job of addressing my concerns and I am content to let it be published now.

Reviewer #3 (Remarks to the Author):

Thank you for the very careful revision of the manuscript! I very much enjoyed reading it and think it is almost ready for publication.

- SI page 8, the sentence "For the overall GDP loss we assume that it only changes the mean but not the shape of the in-come distribution." Is unclear. What is the overall GDP loss? You mean system changes in the economy induced by the carbon price?
- SI Figure 4 shows that the fit is rather weak. Would say this is a shortcoming. Also, how does this fit match the relatively high R2 values in table S1?
- Heading 8.3 is not well chosen "Fit results"
- SI Table 2: Now you call the results logit regression, but be aware you use if I am correct a logit transformation of poverty rate and then a linear regression model. Logit regression only allows a binary dependent variable which poverty rate is not. (eq. 4 in the main text)
- L43: Would say climate policy instead of climate protection strategies
- L407. I don't see the added value of the section Sensitivity to Mitigation Target
- The (new) figure 1b I would suggest to NOT use log scale as it is misleading with poverty headcount in my opinion.

We would like to thank all three reviewers for their constructive comments and their positive feedback on our changes in the first revision. Below we respond to their remaining comments individually.

Reviewer #1 (Remarks to the Author):

The paper, in the revised version, looks extremely improved. Expanding the analysis to consider all SSPs is a value added for the paper and it allows understanding the range of policy outcomes under different socioeconomic development and mitigation efforts in place. Furthermore, the rephrasing of recycling scheme description improves greatly the understanding and the additional “strongly progressive” scheme conveys a strong message to that part of scholars which sees mitigation as necessarily detrimental to poverty reduction.

Many thanks for the positive feedback on our changes in the first revision!

I am still not completely convinced by the rewording of “household income” into “real income”. In Hussein et al. 2013, the use of this term seems more in line with its meaning, i.e. income cleaned by inflationary effects. But, in the paper, the Authors clearly state that they account for both price and quantity change of the consumed commodities. So I still think that “real income” is improperly used here.

Thanks for pointing this out. We replaced the term “real income” with the more precise “income equivalent net of climate policy induced changes” (first occurrences). In subsequent occurrences, we used the shorter version “income equivalent” (same also in the SI). We kept the clarification that this can also be viewed as a welfare change measured in monetary units (added during the first review round). We further added an additional clarification that the price-induced changes in quantities are also considered.

Reviewer #2 (Remarks to the Author):

The authors have done a good job of addressing my concerns and I am content to let it be published now.

Thanks a lot for the positive feedback on the revision!

Reviewer #3 (Remarks to the Author):

Thank you for the very careful revision of the manuscript! I very much enjoyed reading it and think it is almost ready for publication.

Thanks a lot, both for the positive feedback on the revision and the helpful further comments.

- *SI page 8, the sentence “For the overall GDP loss we assume that it only changes the mean but not the shape of the in-come distribution.” Is unclear. What is the overall GDP loss? You mean system changes in the economy induced by the carbon price?*

Yes, that is what we meant. To be clearer, we changed this to: “The loss in GDP caused by mitigation action such as carbon pricing reduces the average income of households. We assume that this loss is distributed in a neutral way across income groups, i.e. that it changes only the mean of the income distribution, but not the level of inequality.” We also rephrased to “GDP loss caused by mitigation action” at another occurrence of “overall GDP loss” (SI Sec. 3.3, p.6).

- *SI Figure 4 shows that the fit is rather weak. Would say this is a shortcoming. Also, how does this fit match the relatively high R2 values in table S1?*

Our regression specification for the energy expenditure share (SI eq. 12) includes country fixed effects (μ_c), which absorb variations in energy expenditure share between countries that are uncorrelated with income (e.g. different climate zones, cultural differences, etc.). In SI Fig. 4 we plot the mean relation; a substantial part of the scatter around it is due to these between-country differences. We had already mentioned this in the discussion around Fig. 4, but now made it more explicit in the figure caption. This also explains the reasonable R^2 despite the large scatter visible in the figure.

Note further that the country fixed effects drop out when calculating the income elasticity (α_{FE} , SI eq. 13). Thus the between-country scatter does not affect our results for α_{FE} (which is what we use in the subsequent analysis) substantially. We also expanded this point (p. 11 of SI). See also the sensitivity test on splitting the expenditure data into country groups with high or low energy taxes or subsidies (p. 11 of SI, directly below).

- *Heading 8.3 is not well chosen “Fit results”*

We changed this to “Model parameters”.

- *SI Table 2: Now you call the results logit regression, but be aware you use if I am correct a logit transformation of poverty rate and then a linear regression model. Logit regression only allows a binary dependent variable which poverty rate is not. (eq. 4 in the main text)*

True, thanks for catching this misnomer. We changed this to simply say “regression model”. We also updated the flowchart (SI Fig. 1) accordingly, and also changed two occurrences of “logistic regression model” to “regression model” (SI right before Eq. 16, Methods of main text, L. 880).

- *L43: Would say climate policy instead of climate protection strategies*

Done.

- *L407. I don't see the added value of the section Sensitivity to Mitigation Target*

The stringency of the climate target determines the ambition and timing of mitigation, which in turn affects (i) the magnitude of the side effect on poverty, and (ii) the magnitude of the carbon pricing revenues, i.e. the ability to compensate the side effect. The combination of

these two effects results in a non-trivial net effect on the poverty headcount: in 2030 more ambitious climate policies can lead to *less* poverty (revenue recycling more than compensates the side effect), whereas for 2050 it is the other way round (1.5°C scenarios reach net-zero around 2050, so there are very little carbon pricing revenues left for redistribution). So we do think that there is added value in discussing this. We now point out more clearly in the section what can be learned from this sensitivity analysis.

- *The (new) figure 1b I would suggest to NOT use log scale as it is misleading with poverty headcount in my opinion.*

Agreed, the log-scale over-emphasized the differences at the lower end, especially between SSP1 and SSP5. We changed Fig. 1b to linear scale (slightly reducing x and y ranges), and removed the reference to the log-scale from the caption. We further slightly expanded the description of the uncertainty bands in the caption (clarifying why the uncertainty bands shrink in size as the headcount gets smaller).

We also changed Fig. 6b (same as Fig. 1b, but for a higher poverty line) accordingly.

For Fig. 3 we prefer to keep the log-scale, as here the aim is to emphasize the difference between baseline and the two policy cases. We highlighted the use of log-scale in the caption of Fig. 3.

Additional minor change (unrelated to reviewer comments):

We updated the flowchart in the SI (Supplementary Figure 1) to reflect the changed naming of the revenue recycling scenarios (neutral / progressive / strongly progressive) from the first review round.

REVIEWERS' COMMENTS:

Reviewer #3 (Remarks to the Author):

Thank you for the further revision. I am happy with the few issues that were addressed appropriately and think the manuscript is ready for publication and an important contribution to the link between climate policy and poverty.

REVIEWERS' COMMENTS:

Reviewer #3 (Remarks to the Author):

Thank you for the further revision. I am happy with the few issues that were addressed appropriately and think the manuscript is ready for publication and an important contribution to the link between climate policy and poverty.

We would like to thank all three reviewers for their helpful and constructive feedback that certainly improved the paper!